# Formation reaction mechanism and infrared spectra of *anti-trans*-methacrolein oxide and its associated precursor and adduct radicals

Jia-Rong Cai[1], Jung-Hsuan Su[1] & Yuan-Pern Lee [ID] [1,2,3 ✉]

Methacrolein oxide (MACRO) is an important carbonyl oxide produced in ozonolysis of isoprene, the most abundantly-emitted non-methane hydrocarbon in the atmosphere. We employed a step-scan Fourier-transform infrared spectrometer to investigate the source reaction of MACRO in laboratories. Upon UV irradiation of precursor $CH_2IC(CH_3)CHI$ **(1)**, the $CH_2C(CH_3)CHI$ radical **(2)** was detected, confirming the fission of the allylic C–I bond rather than the vinylic C–I bond. Upon UV irradiation of **(1)** and $O_2$ near 21 Torr, *anti-trans*-MACRO **(3a)** was observed to have an intense OO-stretching band near $917 \, cm^{-1}$, much greater than those of *syn*-$CH_3CHOO$ and $(CH_3)_2COO$, supporting a stronger O–O bond in MACRO because of resonance stabilization. At increased pressure (86–346 Torr), both reaction adducts $CH_2C(CH_3)CHIOO$ **(4)** and $(CHI)C(CH_3)CH_2OO$ **(5)** radicals were observed, indicating that $O_2$ can add to either carbon of the delocalized propenyl radical moiety of **(2)**. The yield of MACRO is significantly smaller than other carbonyl oxides.

[1] Department of Applied Chemistry and Institute of Molecular Science, National Yang Ming Chiao Tung University, Hsinchu 300093, Taiwan. [2] Center for Emergent Functional Matter Science, National Yang Ming Chiao Tung University, Hsinchu 300093, Taiwan. [3] Institute of Atomic and Molecular Sciences, Academia Sinica, Taipei 106319, Taiwan. ✉email: yplee@nycu.edu.tw

soprene [2-methyl-1,3-butadiene, $CH_2 = CH–C(CH_3) = CH_2$] is the most abundantly emitted non-methane volatile organic compound (VOC) emitted into Earth's atmosphere; an emission budget ~530 Tg year$^{-1}$, ~70% of the total biogenic VOC emission, was estimated[1,2]. Ozone is responsible for the removal of ~10% of isoprene[3,4]. The ozonolysis of isoprene produces three carbonyl oxides (so-called Criegee intermediates); formaldehyde oxide ($CH_2OO$), methyl vinyl ketone oxide [MVKO, $C_2H_3C(CH_3)OO$], and methacrolein oxide [MACRO, $CH_2C(CH_3)CHOO$] are produced with estimated branching ratios 58, 23, and 19%, respectively[3,5,6].

Previously, detecting carbonyl oxides from ozonolysis of alkenes in laboratories was difficult because these source reactions are slow but the carbonyl oxides thus produced are highly reactive. Welz et al. reported an original reaction scheme to generate the simplest carbonyl oxide $CH_2OO$ in laboratories from the reaction of $CH_2I$ with $O_2$ on photolysis of $CH_2I_2$ in $O_2$ with ultraviolet (UV) light[7]. Further extension of this scheme to produce substituted carbonyl oxides has promoted active research, as discussed in several reviews[8–14].

To produce MACRO, $CH_2C(CH_3)CHOO$, following this method from photolysis of $CH_2C(CH_3)CHI_2$ in $O_2$ is, however, difficult because this precursor is extremely unstable. Vansco et al. reported a unique method to produce MACRO on photolysis at 248 nm of a gaseous mixture of 1,3-diiodo-2-methyl-prop-1-ene [$CH_2IC(CH_3)CHI$] (1) and $O_2$ that was pulsed into a quartz capillary reactor tube[15]. These authors assumed that photolysis of $CH_2IC(CH_3)CHI$ (1) at 248 nm resulted in a preferential dissociation of the allylic, rather than the vinylic, C–I bond, to form the iodoalkenyl radical 3-iodo-2-methyl-prop-1-en-3-yl [$CH_2C(CH_3)CHI$] (2). Subsequent addition of $O_2$ with this resonance-stabilized radical (2) to form adduct 3-hydroperoxy-3-iodo-2-methyl-prop-1-ene $CH_2C(CH_3)CHIOO$ (4) that

readily breaks the remaining C–I bond to produce the carbonyl oxide MACRO (3). A detailed reaction scheme appears in Fig. 1; the chemical formula and labels of key species are also presented. Four conformers of MACRO, *anti-trans*, *syn-cis*, *syn-trans*, and *anti-cis* (with increasing energy, shown in Fig. 1) are predicted to exist; *syn-* and *anti-* indicate the orientation of the $CH_2 = C(CH_3)$ moiety relative to the terminal oxygen atom, and *cis-* and *trans-* indicate the relative orientation of the C = C bond and the C = O bond. The UV-visible spectrum of jet-cooled MACRO was obtained by means of UV-visible depletion of the parent ion signal of MACRO at $m/e = 86$ upon photoionization at 10.5 eV[15], but this spectrum with maximum absorption near 380 nm provides no specific information on the conformation of MACRO; equal populations of the conformers were assumed because these four conformers were predicted to have energies within 13.3 kJ mol$^{-1}$ according to the CCSD(T)-F12/CBS(TZ-F12,QZ-F12)// B2PLYP-D3/cc-pVTZ method[15]. Lin et al. reported the direct UV-visible absorption spectrum of MACRO with a maximum at 397 nm upon photoirradiation of the same precursors at 248 nm and 298 K[16]. Unlike MVKO[17], the near-infrared action spectra of MACRO could not be obtained by probing OH radicals because OH is not a significant reaction product. The mechanism for the formation of MACRO, including the characterization of the associated iodoalkenyl radical (2) before its reaction with $O_2$ and the iodoperoxy radical adducts (4) and (5) before the fission of the second C–I bond, has not been identified. The mid-infrared spectrum of MACRO and other related intermediates will provide a clue to the conformation of these species and a detailed mechanism for the production of MACRO from UV photolysis of (1) in $O_2$.

We have previously employed a unique step-scan Fourier-transform infrared (FTIR) absorption technique to detect unstable species[18]. The wide spectral and temporal coverage

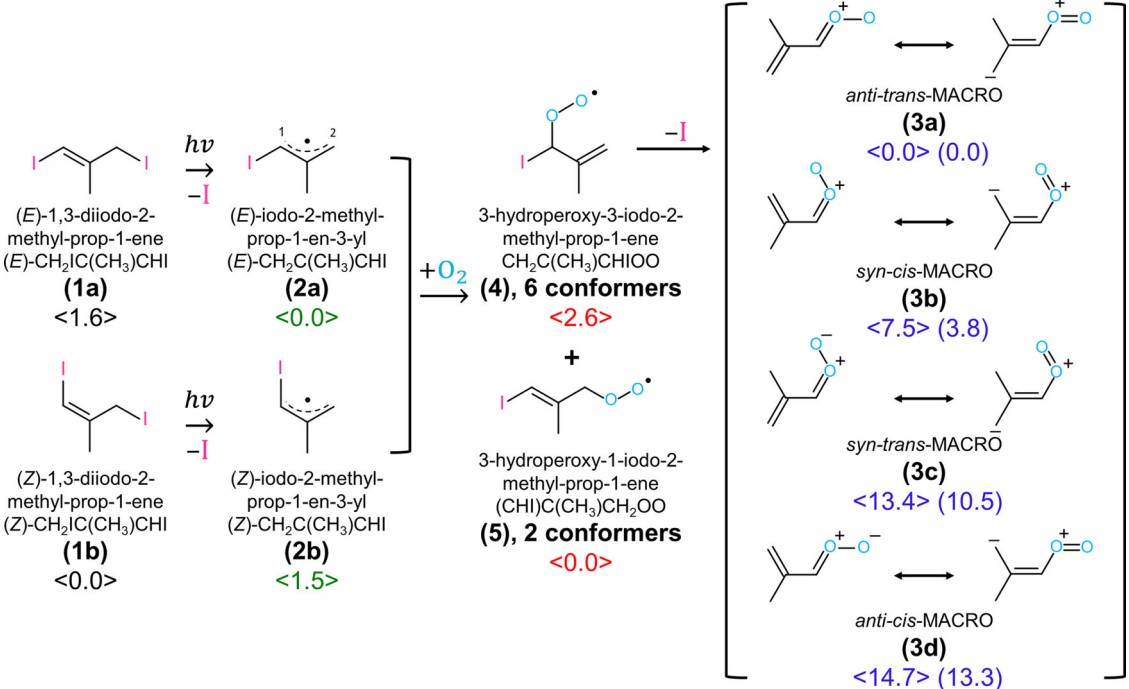

**Fig. 1 Reaction scheme of photolysis of 1,3-diiodo-2-methyl-prop-1-ene (1) to produce methacrolein oxide (MACRO, 3).** Two conformers of 1,3-diiodo-2-methyl-prop-1-ene [$CH_2IC(CH_3)CHI$] (1) and iodo-2-methyl-prop-1-en-3-yl [$CH_2C(CH_3)CHI$] (2), and four conformers of MACRO [$CH_2C(CH_3)CHOO$] (3) are depicted. Iodoperoxy adducts 3-hydroperoxy-3-iodo-2-methyl-prop-1-ene (4), $CH_2C(CH_3)CHIOO$, and 3-hydroperoxy-1-iodo-2-methyl-prop-1-ene (5), $(CHI)C(CH_3)CH_2OO$, have 6 and 2 conformers (not shown), respectively. The major resonance structures of MACRO are also shown. The relative energies (in kJ mol$^{-1}$), computed with the B3LYP/aug-cc-pVTZ-pp method, are shown in brackets for conformers of each species; those of CCSD(T)-F12/ CBS(TZ-F12,QZ-F12)//B2PLYP-D3/cc-pVTZ, reported by Vansco et al.[15] are listed in parentheses.

enables us to monitor several reaction intermediates simultaneously; their temporal behavior provides valuable information to understand the detailed reaction mechanism. With this technique, we have successfully detected infrared spectra of carbonyl oxides $CH_2OO$[19,20], $CH_3CHOO$[21], $(CH_3)_2COO$[22], MVKO[23], and the associated adduct $CH_2IOO$[24]; we explored also the mechanism and the intermediates in reactions of $CH_2OO$ with $CH_2OO$[25], $SO_2$[26], $HC(O)OH$[27], and HCl[28].

In this work, we extended our focus to MACRO and investigated the UV photodissociation of the precursor $CH_2IC(CH_3)CHI$ (1) to observe the $CH_2C(CH_3)CHI$ radical (2), confirming that only the allylic C–I bond was broken. When oxygen at varied pressure was added, the IR spectra of the *anti-trans*-MACRO [$CH_2C(CH_3)CHOO$] (3a) and the adducts $CH_2C(CH_3)CHIOO$ (4) and $(CHI)C(CH_3)CH_2OO$ (5) radicals were characterized. The IR spectrum of MACRO indicates that the preferred conformation is *anti-trans* and provides direct spectral evidence for resonance stabilization and hyper-conjugation of MACRO.

## Results and discussion

**Quantum-chemical calculations**. Although Vansco et al.[15] has reported high-level calculations for conformers of MACRO, we performed calculations at the B3LYP/aug-cc-pVTZ level of theory mainly for predictions of vibrational wavenumbers and IR intensities of various conformers of MACRO and other associated species. As summarized in Supplementary Note 1, the optimized geometries and Cartesian coordinates of conformers of precursors (1), iodoalkenyl radicals $CH_2C(CH_3)CHI$ (2) and $CH_2IC(CH_3)CH$ (6), carbonyl oxides MACRO (3), a possible cyclic peroxide product dioxole from unimolecular isomerization of MACRO[29,30], and iodoperoxy radical adducts (4) and (5) are presented in Supplementary Figs. 1–4 and Supplementary Tables 1–4. Relative energies of conformers are also listed in these figures and those of MACRO are compared with high-level calculations by Vansco et al.[15].

Computed scaled harmonic vibrational wavenumbers and IR intensities of these species are listed in Supplementary Tables 5–9, respectively. The harmonic vibrational wavenumbers of all species discussed in this paper were scaled according to the equation $y = 0.9683\ x + 11.5$, in which $y$ and $x$ are scaled and harmonic vibrational wavenumbers, respectively; this equation was derived on fitting the observed bands of precursor (1a) with computed harmonic vibrational wavenumbers. The average absolute deviation of scaled harmonic vibrational wavenumbers of (1a) from observed wavenumbers is $7.5 \pm 6.4\ cm^{-1}$; the error represents one standard deviation in the fitting. The computed anharmonic vibrational wavenumbers of the four conformers of MACRO, (3a)−(3d), and dioxole are also listed in Supplementary Table S7. The rotational parameters for each vibrational state of (3a)−(3d) are listed in Supplementary Table S10.

**IR spectra of the iodoalkenyl radical $CH_2C(CH_3)CHI$ (2)**. Precursor (1) is predicted to exist in (Z)- and (E)-conformations; the (E)-conformer (1a) is predicted to have energy $1.6\ kJ\ mol^{-1}$ greater than the (Z)-conformer (1b) at the B3LYP/aug-cc-pVTZ level of theory. The IR spectra of (1a) and (1b) are shown in Supplementary Fig. 5 and their wavenumbers are listed in Supplementary Table 5. These spectra and results on photolysis (Fig. 2, same as Supplementary Fig. 6) are discussed in Supplementary Note 2. When precursor (1), a mixture of (1a) and (1b) with spectrum shown in Fig. 2a, was irradiated with light at 248 nm, the intensity of its bands decreased significantly and weak new features appeared (Fig. 2b). Processing these difference spectra (Fig. 2c–e) by adding back the loss of the precursor yielded Fig. 2f–h, as discussed in Supplementary Note 2; the regions in which the absorption of the precursor might interfere are marked with gray rectangles. Three features near

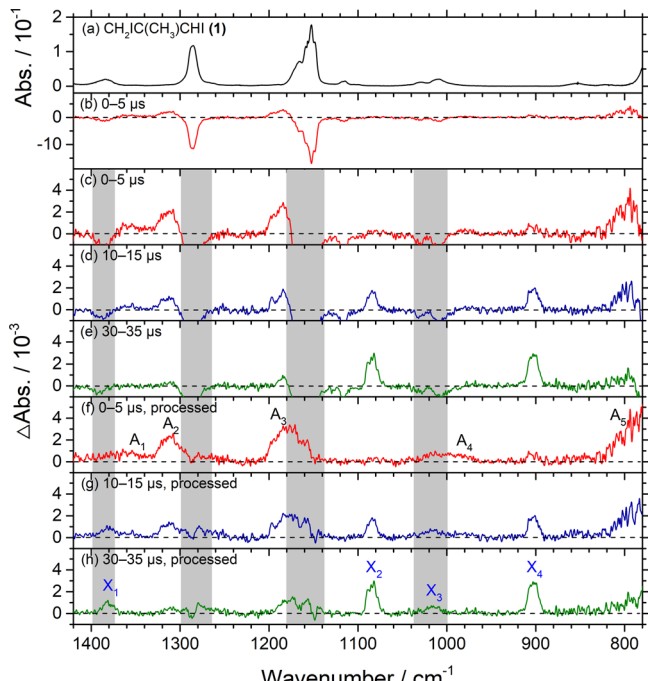

**Fig. 2 Observed and processed spectra in region 1420–780 cm⁻¹ upon photolysis at 248 nm of a flowing mixture of $CH_2IC(CH_3)CHI$ (1)/$N_2$ (0.030/20.0 Torr). a** Absorption spectrum before photolysis. **b** Difference spectra recorded 0–5 μs after photolysis. **c–e** Expanded difference spectra recorded 0–5, 10–15, and 30–35 μs after irradiation; negative bands are truncated. **f–h** Processed spectra of **c–e** with absorption bands of precursor (1), spectrum **a**, added back to eliminate negative bands. Gray areas represent regions of possible interference from absorption of the parent molecule. New features with decreasing intensity are marked A₁-A₅ in **f**. New features with increasing intensity are marked X₁-X₄ in **h**. The spectral resolution is 1.0 cm⁻¹.

1312, 1178, and 778 cm⁻¹ with greater intensities and two weak ones near 1364 and 1002 cm⁻¹ appeared upon UV irradiation and decreased in intensity with reaction period following nearly the same proportions; these transient features in group A, marked as A₁-A₅ in Fig. 2f, are associated with the primary photolysis product. Four features near 1380, 1084, 1014, and 902 cm⁻¹ increased in intensity continuously; these features in group X, marked as X₁-X₄ in Fig. 2h, are associated with the final product.

The spectrum consisting of bands in group A is reproduced in Fig. 3a as the red trace. A corresponding spectrum observed in experiments with (E)-$CH_2IC(CH_3)CHI$ (1a) is shown as a black trace in Fig. 3a; these two traces are similar except that the red trace has a better signal-to-noise ratio (SNR). The stick IR spectra of both conformers of two possible photolysis products, (E)- and (Z)-$CH_2C(CH_3)CHI$, (2a) and (2b), and (E)- and (Z)-$CH_2IC(CH_3)CH$, (6a) and (6b), according to the scaled harmonic vibrational wavenumbers and intensities predicted with the B3LYP method are shown in Fig. 3b–e. The observed new features near 1364, 1312, 1178, 1002, and 778 cm⁻¹ agree best, in terms of wavenumbers and relative intensities, with lines predicted near 1355, 1304, 1187, 1004, and 793 cm⁻¹ for (2a), the conformer with the least energy; a small contribution of (2b) cannot be positively excluded because of the similarity in predicted spectra. A comparison of observed bands with calculations in this spectral region is presented in Table 1. In contrast, the observed features in group A agree poorly with the spectra predicted for (6a) and (6b). This observation of bands of (2) confirms that irradiation of (1) at 248 nm caused the breaking of the allylic C–I bond rather than the vinylic C–I bond. Furthermore, because the spectra observed after photolysis of (1a) and (1) are

similar, we deduced that the conformation of (2) was scrambled upon UV photolysis of (1); the barrier to convert (2b) to (2a) is ~50 kJ mol$^{-1}$, much smaller than the excess energy ~300 kJ mol$^{-1}$ at 248-nm photolysis. The calculated energy difference, ~1.5 kJ mol$^{-1}$, between (2a) and (2b) implies a population ratio 65:35 according to the Boltzmann distribution at 298 K, but the observed spectra seem to show a contribution of (2b) smaller than the predicted ratio if one considers that a doublet of similar intensity was predicted for the (Z)-conformer in region 1300–1350 cm$^{-1}$ (Fig. 3c), but only one significant feature was observed. We are, however, uncertain about this population ratio because the predicted IR intensities might have large errors.

The product of dimerization of iodo-radicals (2) was also observed at a later period. Because of resonance, two forms 1-iodo-2-methyl-prop-1-en-3-yl (2-1) and 3-iodo-2-methyl-prop-1-en-3-yl (2-2) might exist for (2), as shown in Fig. 4a for the E-conformer as an example. Possible secondary reactions are shown in Fig. 4b–e, with diene products (7)−(10). The observed spectrum of the end product, group X, is reproduced in Fig. 5a and compared with predicted spectra of (7)−(10) in Fig. 5b–e. Observed bands in group X near 1380, 1084, 1014, and 902 cm$^{-1}$ agree with scaled harmonic vibrational wavenumbers and relative intensities predicted near 1381, 1079, 1008, and 927 cm$^{-1}$ for 3,4-diiodo-2,5-dimethyl-hexa-1,5-diene (7), produced from dimerization of (2-2), shown in Fig. 4b. The observation of end product (7) further supports that only the allylic C–I bond rather than the vinylic C–I bond was broken upon irradiation at 248 nm, and that the radical thus produced, (2), exists in 3-iodo-2-methyl-prop-1-en-3-yl (2-2) as its major form (in which C1–C4 has a double bond character), consistent with the predicted bond length of C1–C4 (1.380 Å) slightly smaller than that of C1–C2 (1.391 Å) for (2a); C1 is the central carbon atom and C2 is the carbon with the I atom, as labeled in Supplementary Fig. 1c.

**IR spectrum of carbonyl oxide *anti-trans*-MACRO (3a).** Results and data processing of photolysis of flowing mixtures of (1) (30 mTorr) and O$_2$ (20.0–21.0 Torr), recorded with external and internal ADC (with improved SNR), are presented in Supplementary Figs. 7 and 8 (reproduced as Fig. 6), respectively, and discussed in detail in Supplementary Note 3. Processing observed difference spectra recorded with an internal ADC (Fig. 6a–c) by removing contributions of (2) and end product methacrolein and adding back the loss of the precursor yielded Fig. 6d–f; the regions in which the absorption of the precursor might interfere are marked with gray rectangles. Further processing by taking out contributions of other stable products (Fig. 6f) produced cleaner spectra, as shown in Fig. 6g, h. Only one prominent band near 917 cm$^{-1}$ showed a transient nature and attained its maximum 5–10 μs after irradiation; we mark this as B$_1$ in Fig. 6h. Weak features B$_2$–B$_4$ near 1025, 1332, and 1386 cm$^{-1}$ might also belong to

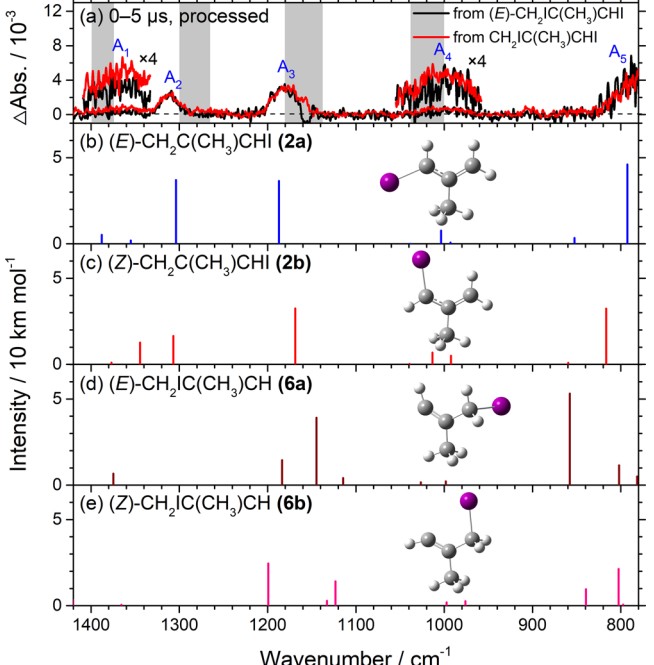

**Fig. 3 Comparison of bands in group A with IR stick spectra of four isomers of iodoalkenyl radicals. a** Spectra of bands in group A recorded 0–5 μs after photolysis of (E)-CH$_2$IC(CH$_3$)CHI (1a) (black) and of a ~1:1 mixture of (E)- and (Z)-CH$_2$IC(CH$_3$)CHI (1) (red, taken from Fig. 2f; bands are labeled A$_1$–A$_5$. Weak bands A$_1$ and A$_4$ are expanded and shifted for clarity. Gray areas represent regions of possible interference from absorption of the parent molecule. IR stick spectra predicted for **b** (E)-CH$_2$C(CH$_3$)CHI (2a), **c** (Z)-CH$_2$C(CH$_3$)CHI (2b), **d** (E)-CH$_2$IC(CH$_3$)CH (6a), and **e** (Z)-CH$_2$IC(CH$_3$)CH (6b) according to scaled harmonic vibrational wavenumbers and IR intensities predicted with the B3LYP/aug-cc-pVTZ-pp method are shown.

**Table 1 Comparison of observed vibrational wavenumbers and IR intensities of (E)-CH$_2$C(CH$_3$)CHI (2a) in region 750–1500 cm$^{-1}$ with those calculated with the B3LYP/aug-cc-pVTZ-pp method.**

| Mode[a] | Sym. | Experiment | | Calculation | | Mode description[e] |
|---|---|---|---|---|---|---|
| | | $\nu$/cm$^{-1}$ | Intensity[b] | $\nu$/cm$^{-1c}$ | Intensity[d] | |
| $\nu_8$ | a′ | | | 1458 | 21.2 | CH$_3$ def. |
| $\nu_9$ | a″ | | | 1448 | 11.1 | CH$_3$ def. |
| $\nu_{10}$ | a′ | | | 1388 | 5.2 | CH$_3$ umbrella |
| $\nu_{11}$ | a′ | 1364 | 28 | 1355 | 2.0 | C$^{(1)}$C$^{(3)}$ str. |
| $\nu_{12}$ | a′ | 1312 | 43 | 1304 | 37.1 | C$^{(1)}$C$^{(2)}$C$^{(4)}$ asym. str. |
| $\nu_{13}$ | a′ | 1178 | 81 | 1187 | 36.5 | CHI bend |
| $\nu_{14}$ | a″ | | | 1039 | 0.0 | C$^{(3)}$H$^{(3)}$H$^{(4)}$ twist |
| $\nu_{15}$ | a′ | 1002 | 33 | 1004 | 7.7 | C$^{(3)}$H$^{(3)}$H$^{(4)}$ wag |
| $\nu_{16}$ | a′ | | | 993 | 0.8 | CH$_2$ ip bend |
| $\nu_{17}$ | a′ | | | 853 | 3.4 | C$^{(1)}$C$^{(3)}$ str./C$^{(1)}$C$^{(2)}$C$^{(4)}$ bend |
| $\nu_{18}$ | a″ | 778 | 100 | 793 | 46.1 | CH$_2$ wag |

[a]Mode numbers are ordered without consideration of symmetry to conform with those of (Z)-CH$_2$C(CH$_3$)CHI (2b).
[b]Percentage of IR intensity relative to the most intense band near 778 cm$^{-1}$.
[c]Harmonic vibrational wavenumber x scaled according to 0.9683 x + 11.5; see text.
[d]In unit km mol$^{-1}$.
[e]Approximate mode description. asym.: *anti*-symmetric; str.: stretch; def.: deform; ip: in-plane.

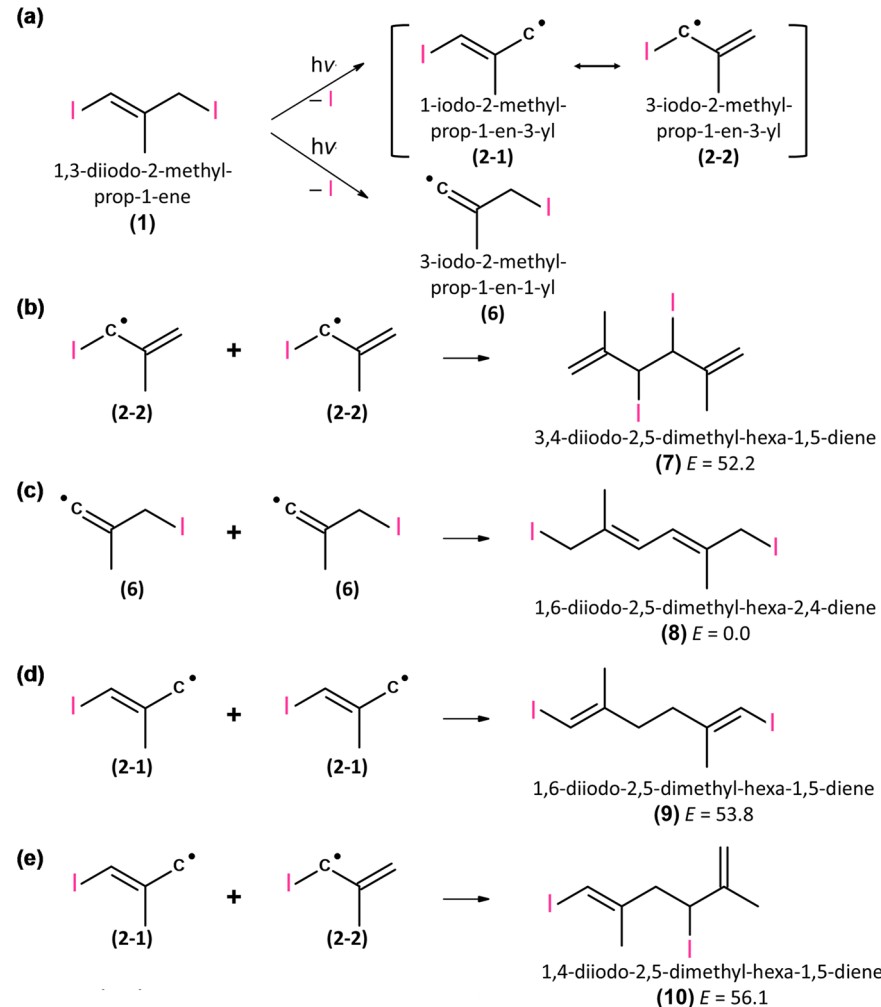

**Fig. 4 Possible dimeric products produced on photolysis of 1,3-diiodo-2-methyl-prop-1-ene (1). a** Mechanism of photolysis of 1,3-diiodo-2-methyl-prop-1-ene **(1)** to produce 1-iodo-2-methyl- prop-1-en-3-yl **(2-1)**, 3-iodo-2-methyl-prop-1-en-3-yl **(2-2)**, and 3-iodo-2-methyl-prop-1-en-1-yl **(6)**. **b–e** Possible dimerization reactions among **(2-1)**, **(2-2)**, and **(6)**; the relative energies (kJ mol⁻¹) of **(7)**−**(10)** are computed with the B3LYP/aug-cc-pVTZ-pp method.

this group; as we are uncertain about these bands because of their small intensities and possible interference from absorption of the parent or the stable products, we indicate them with ? marks.

The photolysis of a mixture of precursor **(1)** and $O_2$ at 248 nm is expected to produce carbonyl oxide MACRO **(3)**, as reported by Vansco et al.[15] The spectrum of group B in Fig. 6h is reproduced in Fig. 7a to compare with the simulated spectra of four conformers of the carbonyl oxides MACRO: *anti-trans-*, *syn-cis-*, *syn-trans-*, and *anti-cis-*$CH_2C(CH_3)CHOO$, **(3a)**–**(3d)**, in Fig. 7b–e. We employed anharmonic vibrational wavenumbers (Supplementary Table 7) and rotational parameters (Supplementary Table 10) predicted with the B3LYP method and simulated the rotational contours with the PGOPHER program (Supplementary Figs. 9 and 10)[31], as described in detail in the Supplementary Note 4. The spectrum of a possible product dioxole simulated according to anharmonic vibrational calculations is also shown in Fig. 7f for comparison.

Observed intense band $B_1$ near 917 cm⁻¹ matches best with the spectrum simulated for *anti-trans-*MACRO **(3a)** in terms of vibrational wavenumbers and relative intensities. The OO-stretching ($v_{15}$) band of **(3a)** was predicted at 944 cm⁻¹ and to have the largest IR intensity (201 km mol⁻¹); other bands in region 850–1450 cm⁻¹ have IR intensities less than 26 km mol⁻¹. For the four conformers, only **(3a)** was predicted to have a single

intense band in this region; others were predicted to have more than two bands with comparable intensities. Furthermore, two *c*-type bands were predicted for modes $v_{24}$ and $v_{25}$ of **(3a)** near 950 and 924 cm⁻¹; they might correspond to the small narrow features near 921 and 902 cm⁻¹, indicated with arrows in Fig. 7a; other conformers were predicted to have only one *c*-type band near 900 cm⁻¹. A comparison of observed bands with predicted vibrational wavenumbers and intensities of **(3a)** is listed in Table 2. The observed band near 917 cm⁻¹ appears to have an insignificant Q-branch and a broader rotational contour as compared with that simulated with the PGOPHER program, presumably because of the contribution of hot bands from excited states of the low-energy vibrational modes. Similar broadening was observed for carbonyl oxides containing a methyl rotor, $CH_3CHOO$[21], $(CH_3)_2COO$[22], and MVKO[23].

A small contribution of *syn-cis-*MACRO **(3b)**, the second least-energy conformer with energy 7.5 (3.8 from CCSD(T)-F12[15]) kJ mol⁻¹ greater than *anti-trans-*MACRO **(3a)**, might be assigned to the observed weak bands $B_2$–$B_4$, but we are unable to confirm this definitively because of the small intensity and interference. The observed bands near 1025, 1332, and 1386 cm⁻¹ agree satisfactorily with the more intense bands of **(3b)** predicted near 1030, 1346, and 1383 cm⁻¹. The band predicted near 907 cm⁻¹ might overlap with band $B_1$ of **(3a)**; two bands predicted near

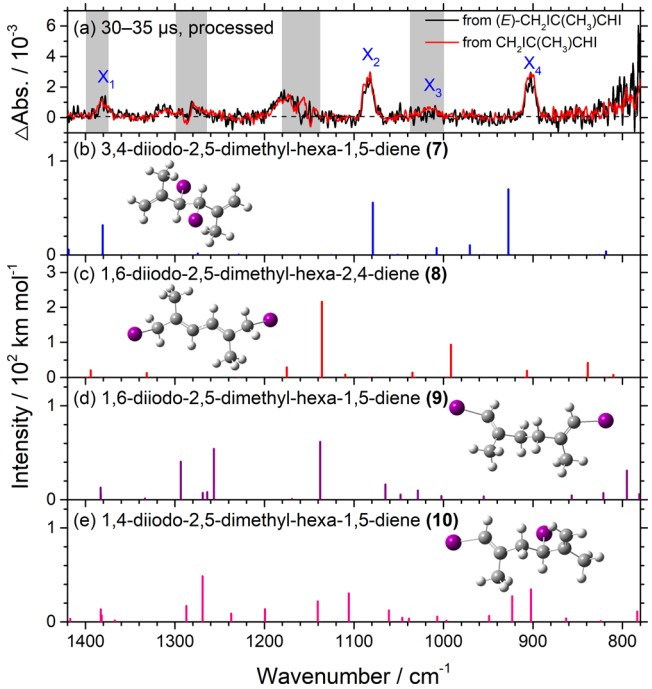

**Fig. 5 Comparison of IR spectra of end products (group X) on photolysis of CH$_2$IC(CH$_3$)CHI (1)/N$_2$ with predicted IR stick spectra of possible products. a** Spectrum of bands in group X recorded 30–35 μs after photolysis of (*E*)-CH$_2$IC(CH$_3$)CHI (black) or a mixture of (*E*)-and (*Z*)-CH$_2$IC(CH$_3$)CHI (red); taken from Fig. 2h; the resolution is 1.0 cm$^{-1}$. Gray areas represent regions of possible interference from absorption of the parent molecule. IR stick spectra according to scaled harmonic vibrational wavenumbers and IR intensities predicted with the B3LYP/aug-cc-pVTZ-pp method are shown for four possible dimers: **b** 3,4-diiodo-2,5- dimethyl-hexa-1,5-diene (**7**), **c** 1,6-diiodo-2,5-dimethyl-hexa-2,4-diene (**8**), **d** 1,6-diiodo- 2,5-dimethyl-hexa-1,5-diene (**9**), and **e** 1,4-diiodo-2,5-dimethyl-hexa-1,5-diene (**10**); the structures are shown in Fig. 4.

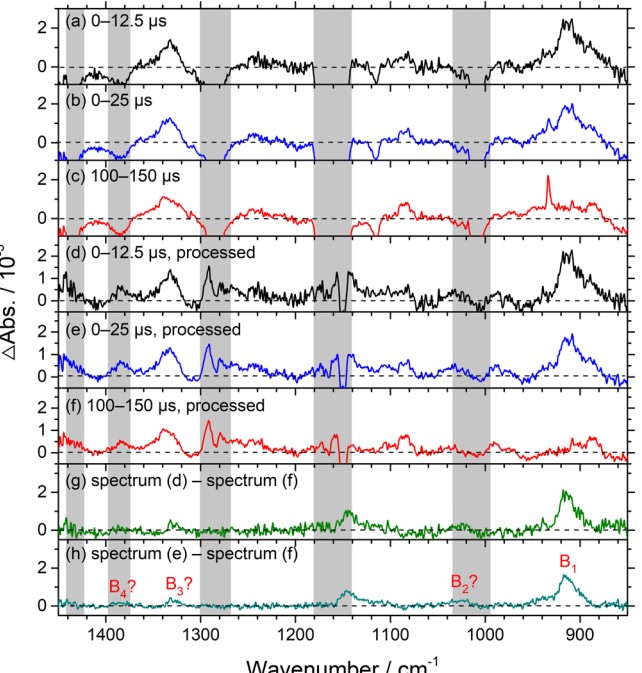

**Fig. 6 Observed and processed spectra in region 1450–850 cm$^{-1}$ upon photolysis at 248 nm of a flowing mixture of CH$_2$IC(CH$_3$)CHI (1)/O$_2$ (0.030/20.0 Torr).** Difference spectra recorded 0–12.5 (**a**), 0–25 (**b**), and 100–150 (**c**) μs after irradiation; negative bands are truncated. **d**–**f** Processed spectra of **a**–**c** with bands of CH$_2$C(CH$_3$)CHI (**2**) and methacrolein (MACR) removed and those of the precursor (**1**) added back. Gray areas represent regions of possible interference from absorption of the parent molecules. **g** Spectrum **d** subtracts spectrum **f**. **h** Spectrum **e** subtracts spectrum **f**. New features are marked B$_1$–B$_4$ in **h**; the latter three are uncertain because of their small intensities. The spectral resolution is 1.0 cm$^{-1}$.

980 cm$^{-1}$ might suffer interference from the parent absorption, so they are unobserved.

The agreement of observed bands with those predicted for the *syn-trans*- and *anti-cis*-MACRO, (**3c**) and (**3d**), is less satisfactory. Considering the computed relative energies of these two conformers, 10.5 (13.4) and 13.3 (14.7) kJ mol$^{-1}$ above (**3a**), our observation of predominant *anti-trans*-MACRO (**3a**) with a negligibly small contribution of (**3c**) and (**3d**) is reasonable; the listed energies are from CCSD(T)-F12[15] and B3LYP (listed parenthetically). The observed spectrum agrees poorly with that predicted for dioxole (Fig. 7f).

**Resonance stabilization of MVKO and MACRO (3).** We compared the lengths of bonds O-O and C-O and the OO-stretching vibrational wavenumbers of (**3a**) with *syn-trans*-MVKO, *syn*-CH$_3$CHOO, *anti*-CH$_3$CHOO, and (CH$_3$)$_2$COO in Table 3 to confirm the resonance stabilization of MVKO and MACRO. The O-O lengths 1.365 Å for (**3a**) and 1.353 Å for *syn-trans*-MVKO are significantly smaller than those of other carbonyl oxides (~1.380 Å). The observed OO-stretching vibrational wavenumbers 917 cm$^{-1}$ for (**3a**) and 948 cm$^{-1}$ for *syn-trans*-MVKO are much greater than the corresponding values 871–887 cm$^{-1}$ for other species. All this evidence supports that the COO moieties of MACRO and MVKO are resonance stabilized by the adjacent vinyl group so that the extended π-electron delocalization strengthens the O-O bond significantly; the major

resonance structures of MACRO and MVKO are shown in Supplementary Fig. 11a and b, respectively; the major resonance structures for four conformers of MACRO are also depicted in Fig. 1. Furthermore, the molecular orbitals of MACRO and MVKO also show delocalization of π-electron densities over the CCCOO skeleton, as shown in Supplementary Fig. 11c for node = 0–2; a more complete set of molecular orbitals have been reported by Vansco et al.[15].

That the observed vibrational wavenumber 917 cm$^{-1}$ for the OO-stretching mode of (**3a**) is smaller than a value, 948 cm$^{-1}$, observed for MVKO can be explained with a concept of hyper-conjugation generally employed in organic chemistry; this resonance structure is also shown in Supplementary Fig. 11a. In this hyper-conjugation structure, the O–O bond has single-bond character, whereas the adjacent C=O bond has double-bond character; this contribution explains that MACRO has a longer O–O bond length and a shorter C–O length, with $R_{OO}$ = 1.365 Å and $R_{CO}$ = 1.266 Å for MACRO and $R_{OO}$ = 1.353 Å and $R_{CO}$ = 1.297 Å for MVKO. Our observation of the OO-stretching wavenumbers of carbonyl oxides provided a direct spectral confirmation of the resonance stabilization and hyper-conjugation of MACRO. Because of this resonance stabilization, a smaller reactivity of MACRO and MVKO was predicted and observed[16,32,33].

**Assignments of iodoperoxy radical adducts.** Results on photolysis of a flowing mixture of (**1**) (60 mTorr) and O$_2$ (21–346 Torr)

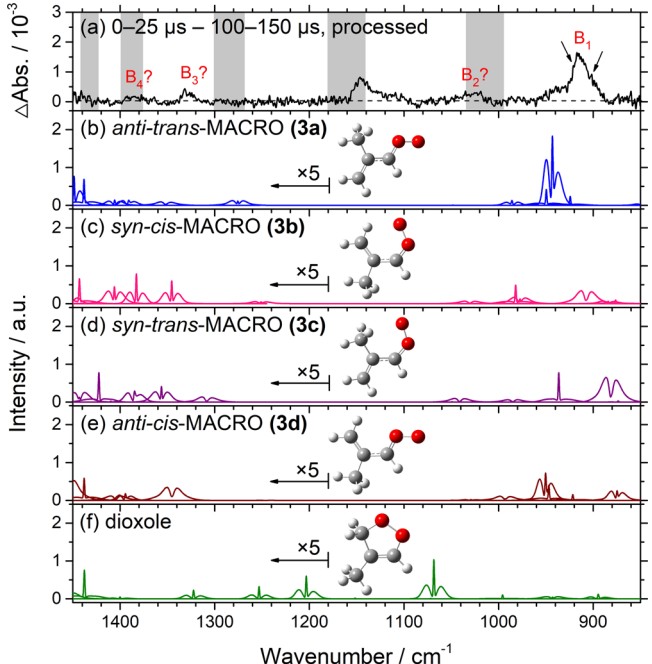

**Fig. 7 Comparison of bands in group B with simulated spectra of various conformers of carbonyl oxide MACRO and dioxole. a** Spectrum of bands in group B after photolysis; taken from Fig. 6h. The spectral resolution is 1.0 cm$^{-1}$. Gray areas represent regions of possible interference from absorption of the parent molecule. IR spectra simulated with PGOPHER ($J_{max} = 150$, $T = 298$ K, fwhm = 1.28 cm$^{-1}$) are shown for **b** anti-trans-MACRO **(3a)**, **c** syn-cis-MACRO **(3b)**, **d** syn-trans-MACRO **(3c)**, **e** anti-cis-MACRO **(3d)**, and **f** dioxole. Band intensities are multiplied by 5 in region 1180–1450 cm$^{-1}$.

are discussed in detail in Supplementary Note 5. Figure 8 (same as Supplementary Fig. 12) shows the results of experiments on $CH_2IC(CH_3)CHI$ **(1)**/$O_2$ (0.060/334 Torr) in a flowing mixture. Details in processing observed difference spectra (Fig. 8b–d) by stripping bands of MACRO and methacrolein and adding back the decrease in precursor bands to obtain Fig. 8g–i are discussed in Supplementary Note 5; the regions in which the absorption of the precursor might interfere are marked with gray rectangles. Two sets of new bands were observed. Bands of group D near 1333, 1243, 990, and 886 cm$^{-1}$, marked D$_1$–D$_4$ in Fig. 8i, are associated with a more stable intermediate; bands of group C near 1116, 1031, 914, and 888 cm$^{-1}$, marked C$_1$–C$_4$ in Fig. 8j, are associated with a less stable intermediate, according to their temporal profiles.

For the simplest carbonyl oxide $CH_2OO$, both $CH_2OO$ and the adduct $ICH_2OO$ were produced from UV photolysis of $CH_2I_2$ + $O_2$; the yield of $ICH_2OO$ increased with pressure because the adduct was stabilized at high pressure[24]. Similarly, the adduct $C_2H_3C(CH_3)IOO$ was produced on photolysis of a mixture of the precursor of MVKO, $(CH_2I)HC = C(CH_3)I$, and $O_2$ at 248 nm at high pressure[23]. As discussed previously, photolysis of precursor **(1)** produces only $CH_2C(CH_3)CHI$ **(2)**, not $CH_2IC(CH_3)CH$ **(6)**; possible structures of the adducts are hence $CH_2C(CH_3)CHIOO$ **(4)** or $(CHI)C(CH_3)CH_2OO$ **(5)**, with $O_2$ added to the carbon atom on either side of the delocalized propenyl radical moiety.

Six conformers exist for **(4)** (Supplementary Fig. 3). The conformer of least energy is designated **(4a)**. Only conformer **(4b)** has energy within 3 kJ mol$^{-1}$ of **(4a)**; the other four conformers have energies 9–16 kJ mol$^{-1}$ greater than **(4a)**. Two conformers exist for $(CHI)C(CH_3)CH_2OO$ **(5)** (Supplementary Fig. 4).

Conformer **(5b)** has energy 0.6 kJ mol$^{-1}$ greater than **(5a)**. The energy of **(4a)** is greater than that of **(5a)** by 2.6 kJ mol$^{-1}$.

The spectra of bands in groups C and D are reproduced in Fig. 9a, d, respectively. These spectra are compared with the predicted stick spectra of the two least-energy conformers $CH_2C(CH_3)CHIOO$, **(4a)** and **(4b)**, in Fig. 9b, c, respectively, and the two conformers of $(CHI)C(CH_3)CH_2OO$, **(5a)** and **(5b)**, and dioxole in Fig. 9e–g, respectively. Observed bands C$_1$–C$_4$ near 1116, 1031, 914, and 888 cm$^{-1}$ agree satisfactorily with the scaled harmonic vibrational wavenumbers predicted for the four most intense bands of **(4a)**, near 1135, 1022, 937, and 902 cm$^{-1}$ in this region (Table 4). We could not positively exclude a possible contribution of **(4b)** to the observed spectrum because of the similarity in predicted spectra. The other four conformers **(4c)** −**(4f)** are expected to have insignificant contributions because of greater energy.

Observed bands D$_1$–D$_4$ near 1333, 1243, 990, and 886 cm$^{-1}$ agree satisfactorily with the four most intense bands of $(CHI)C(CH_3)CH_2OO$ **(5a)** predicted near 1332, 1280, 1034, and 889 cm$^{-1}$ in this region (Table 4). Similarly, we could not exclude positively a possible contribution of **(5b)** to the observed spectrum of group D because of the similarity in predicted spectra. Neither spectrum of group C or D shows a satisfactory agreement with that predicted for dioxole, even though we cannot exclude the possibility that dioxole might contribute to part of band C$_1$.

Adducts **(4)** and **(5)** correspond to the addition of $O_2$ to either carbon of the delocalized propenyl radical moiety of **(2)**; this is the first case that both adducts were observed. Because the C–I bond in **(4)** is allylic, whereas that in **(5)** is vinylic, we expect that the dissociation of the C–I bond of **(5)** has a greater barrier; **(5)** is hence expected to be more stable, as was observed experimentally.

**Relative yields of (3), (4), and (5) as a function of pressure**. We performed similar experiments with $O_2$ at pressures near 21, 86, 229, and 346 Torr and recorded the spectra with an external digitizer. The processed spectra for 0–5 and 30–35 μs after photolysis are compared in Supplementary Fig. 13. We estimated the relative yields of **(3)**, **(4)**, and **(5)** at varied pressures using two methods, as discussed in Supplementary Note 6 and summarized in Supplementary Table 11. The relative variations in intensities derived from spectral stripping factors for bands in groups C and D (method I) are more reliable. For **(3)**, the ratio relative to the experiment at 21.0 Torr remains similar (1.29–1.38) at 86–346 Torr. For **(4)** and **(5)**, the ratios increase with pressure, with ratios 1.91 and 1.62 at 346 Torr, respectively; **(4)** increases more than **(5)**. Unlike what was observed for $CH_2OO$[24] and MVKO[23], our experiments show no significant increase of **(4)** at the expense of **(3)** as pressure increases. Furthermore, even though the estimate might have large errors due to uncertainties in predicted IR intensities, the yield of MACRO **(3a)** from photolysis of the precursor is significantly smaller than other carbonyl oxides, in the range 6–8% as listed in Supplementary Table 11.

We calculated the potential-energy scheme for the source reaction with the CCSD(T)//B3LYP/aug-cc-pVTZ-pp method, as shown in Fig. 10. The formation of adducts **(4a)** and **(5a)** from **(2a)** + $O_2$ is exothermic by ~73 and 75 kJ mol$^{-1}$, respectively, whereas the formation of **(3a)** + I is endothermic by ~10 kJ mol$^{-1}$. This endothermicity might explain that MACRO was produced with a yield significantly smaller than other carbonyl oxides because other reactions are slightly exothermic. The smaller pressure effect for **(5)** than for **(4)** is consistent with a larger barrier for the C–I bond fission of **(5)**. It is unclear, however, why the yield of MACRO did not decrease for pressure above 21 Torr. One possibility is that $O_2$ serves as a reactor instead of a quencher and

**Table 2 Comparison of observed vibrational wavenumbers and IR intensities of *anti-trans*-CH$_2$C(CH$_3$)CHOO (3a) and *syn-cis*-CH$_2$C(CH$_3$)CHOO (3b) in region 840–1410 cm$^{-1}$ with those calculated with the B3LYP/aug-cc-pVTZ method.**

| Mode | Sym. | Experiment | | Harmonic | | Anharmonic | Mode description[d] |
|------|------|------------|------|----------|------|------------|---------------------|
| | | $\nu$/cm$^{-1}$ | Int.[a] | $\nu$/cm$^{-1b}$ | Int.[c] | $\nu$/cm$^{-1}$ | |
| *anti-trans*-CH$_2$C(CH$_3$)CHOO **(3a)** | | | | | | | |
| $v_9$ | a′ | | | 1408 | 4.8 | 1406 | CH$_2$ bend |
| $v_{10}$ | a′ | | | 1391 | 4.5 | 1392 | CH$_3$ umbrella |
| $v_{11}$ | a′ | | | 1356 | 4.4 | 1352 | C$^{(3)}$C$^{(1)}$C$^{(2)}$ asym. str. |
| $v_{12}$ | a′ | | | 1273 | 5.8 | 1276 | CH *ip* bend |
| $v_{13}$ | a′ | | | 1033 | 0.3 | 1034 | C$^{(3)}$H$^{(3)}$H$^{(4)}$ wag |
| $v_{14}$ | a′ | | | 983 | 16.7 | 985 | CH$_2$ rock |
| $v_{15}$ | a′ | 917 | 100 | 946 | 200.6 | 944 | OO str. |
| $v_{16}$ | a′ | | | 848 | 7.4 | 848 | C$^{(1)}$C$^{(3)}$ str. |
| $v_{23}$ | a″ | | | 1052 | 0.4 | 1048 | C$^{(3)}$H$^{(3)}$H$^{(4)}$ twist |
| $v_{24}$ | a″ | 921? | e | 950 | 26.2 | 950 | CH *oop* bend |
| $v_{25}$ | a″ | 902? | e | 930 | 14.7 | 924 | CH$_2$ wag/CH *oop* bend |
| *syn-cis*-CH$_2$C(CH$_3$)CHOO **(3b)** | | | | | | | |
| $v_9$ | a′ | | | 1396 | 15.0 | 1406 | CH$_3$ umbrella/ CH$_2$ bend |
| $v_{10}$ | a′ | 1386? | 50 | 1380 | 12.4 | 1383 | CH$_3$ umbrella/ CH$_2$ bend |
| $v_{11}$ | a′ | 1332? | 60 | 1350 | 12.4 | 1346 | CH *ip* bend |
| $v_{12}$ | a′ | | | 1247 | 2.3 | 1252 | C$^{(3)}$C$^{(1)}$C$^{(2)}$ asym. str. |
| $v_{13}$ | a′ | 1025? | 100 | 1022 | 15.5 | 1030 | CH$_2$ rock/ C$^{(3)}$H$^{(3)}$H$^{(4)}$ wag |
| $v_{14}$ | a′ | | | 986 | 38.1 | 978 | C$^{(3)}$H$^{(3)}$H$^{(4)}$ wag |
| $v_{15}$ | a′ | 917? | e | 910 | 81.0 | 907 | OO str. |
| $v_{16}$ | a′ | | | 884 | 14.0 | 884 | C$^{(1)}$C$^{(3)}$ str./OO str. |
| $v_{23}$ | a″ | | | 1056 | 0.4 | 1053 | C$^{(3)}$H$^{(3)}$H$^{(4)}$ twist |
| $v_{24}$ | a″ | | | 988 | 26.7 | 982 | CH$_2$ *oop* bend |
| $v_{25}$ | a″ | | | 893 | 5.2 | 876 | CH *oop* bend |

[a]Percentage of IR intensity relative to the most intense band near 917 cm$^{-1}$.
[b]Harmonic vibrational wavenumber $x$ scaled according to 0.9683 $x$ + 11.5; see text.
[c]In unit km mol$^{-1}$.
[d]Approximate mode description. sym.: symmetric; asym.: *anti*-symmetric; str.: stretch; def.: deform; *ip*: in-plane; *oop*: out-of-plane.
[e]Overlap with the band at 917 cm$^{-1}$.

**Table 3 Comparison of O–O and C–O bond lengths and OO-stretching vibrational wavenumbers of Criegee intermediates.**

| | *anti-trans*-MACRO | *syn-trans*-MVKO | *syn*-CH$_3$CHOO | *anti*-CH$_3$CHOO | (CH$_3$)$_2$COO |
|------|--------------------|--------------------|--------------------|--------------------|-----------------|
| r(O–O)/Å | 1.365 | 1.353 | 1.380 | 1.381 | 1.380 |
| r(C–O)/Å | 1.266 | 1.297 | 1.284 | 1.279 | 1.270 |
| $\nu$(OO)/cm$^{-1}$ | 917 | 948 | 871 | 884 | 887 |
| Calculation method | B3LYP/aug-cc-pVTZ | CCSD(T)/cc-pVTZ | NEVPT2(8,8)/aug-cc-pVDZ | NEVPT2(8,8)/aug-cc-pVDZ | B3LYP/aug-cc-pVTZ |
| Reference | This work | 22 | 26 | 26 | 27 |

some decomposition occurs at 21 Torr. Further investigations are needed to clarify this.

## Conclusion

In the absence of O$_2$, upon irradiation of gaseous precursor 1,3-diiodo-2-methyl-prop-1-ene (**1**), CH$_2$IC(CH$_3$)CHI, at 248 nm, iodoalkenyl radical CH$_2$C(CH$_3$)CHI (**2**) was produced, as characterized with bands near 1364, 1312, 1178, 1002, and 778 cm$^{-1}$. This result provided direct spectral evidence to confirm that only the allylic C–I bond, not the vinylic C–I bond, dissociated upon photolysis through observation of only radical product (**2**), not (**6**), and only end product 3,4-diiodo-2,5-dimethyl-hexa-1,5-diene (**7**), produced from the dimerization reaction of (**2**).

When O$_2$ at 21.0 Torr was added to the system, we report the identification of *anti-trans*-CH$_2$C(CH$_3$)CHOO (**3a**) with its intense infrared OO-stretching band near 917 cm$^{-1}$, whereas some weaker bands might be tentatively attributed to the *syn-cis* conformer (**3b**). The observation of a significantly larger OO-stretching wavenumber provides also direct spectral support for a resonance stabilization of MACRO; a OO-stretching wavenumber

smaller than that of its isomer MVKO is associated with the hyper-conjugation of MACRO that weakens the O–O bond slightly.

With O$_2$ at greater pressure, iodoperoxy radical adducts 3-hydroperoxy-3-iodo-2-methyl-prop-1-ene (**4**), CH$_2$C(CH$_3$)CHIOO, characterized with infrared absorption bands at 1116, 1031, 914, and 888 cm$^{-1}$, and 3-hydroperoxy-1-iodo-2-methyl-prop-1-ene (**5**), (CHI)C(CH$_3$)CH$_2$OO, characterized with infrared absorption bands at 1333, 1243, 990, and 886 cm$^{-1}$, were observed. This spectral evidence shows O$_2$ can add to either end of the propenyl radical moiety in (**2**). As pressure increases, the yield of (**3**) remained small, whereas those of (**4**) and (**5**) increased; the enhancement with pressure was more for (**4**), consistent with the expectation that (**5**) has a much larger barrier for the C–I fission. The much smaller yield of (**3**) than for other carbonyl oxides produced from similar reaction schemes is explained by a small endothermicity for the formation of (**3**) + I from (**2**) + O$_2$; formation reactions of other carbonyl oxides are slightly exothermic.

The IR spectra of these four intermediates (**2**)−(**5**) are new; they provide valuable information to probe the associated species

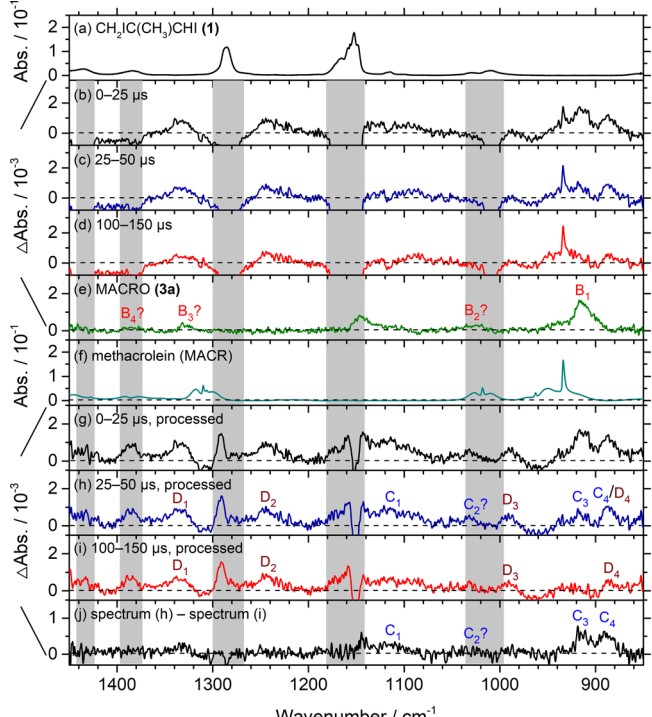

**Fig. 8 Observed and processed spectra in region 1450–850 cm⁻¹ upon photolysis at 248 nm of a flowing mixture of CH₂IC(CH₃)CHI (1)/O₂ (0.060/334 Torr).** a Absorption spectrum before photolysis. Difference spectra recorded 0–25 (**b**), 25–50 (**c**), and 100–150 (**d**) µs after irradiation. **e** Spectrum of MACRO (**3**) taken from Fig. 6h. **f** Absorption spectrum of methacrolein (MACR). **g–i** Processed spectra of **b–d** with bands of CH₂C(CH₃)CHI (**2**) and MACR removed and those of the precursor (**1**) added back. **j** Spectrum **h** subtracts spectrum **i** to remove the contribution of bands in group D. Gray areas represent regions of possible interference from absorption of the parent molecules (**1**). New features are marked C₁–C₄ and D₁–D₄ in **h–j**. The spectral resolution is 1.0 cm⁻¹.

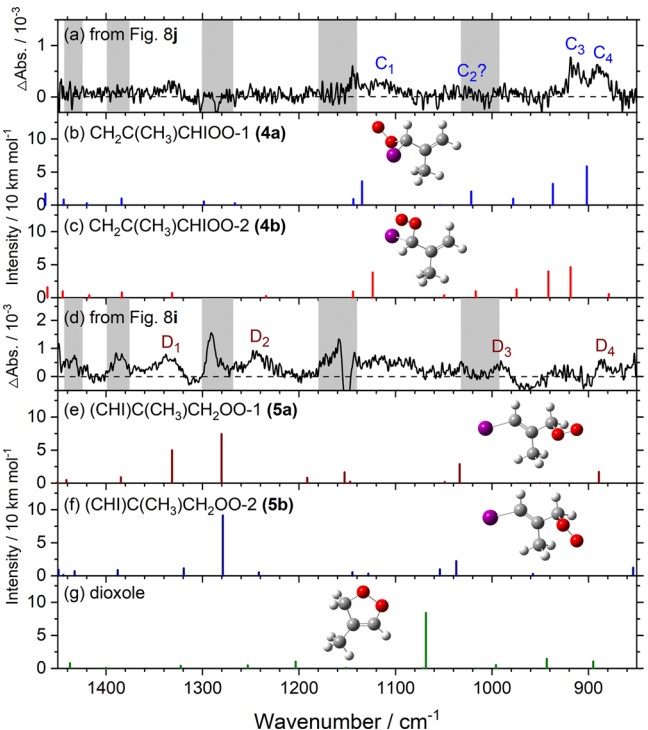

**Fig. 9 Comparison of bands in groups C and D with predicted IR stick spectra of representative isomers of iodoperoxy radicals and dioxole.** a Experimental spectrum of group C, taken from Fig. 8j; bands are labeled C₁–C₄. Gray areas represent regions of possible interference from absorption of the precursor. **d** Experimental spectrum of group D, taken from Fig.8i; bands are labeled D₁–D₄. IR stick spectra according to scaled harmonic vibrational wavenumbers and IR intensities predicted with the B3LYP/aug-cc-pVTZ-pp method are shown for two least-energy conformers of CH₂C(CH₃)CHIOO, (**4a**) and (**4b**), in **b**, **c**, both conformers of (CHI)C(CH₃)CH₂OO, (**5a**) and (**5b**), in **e**, **f**, and dioxole, in **g**.

to understand the mechanism for the formation of carbonyl oxide MACRO from the source reaction. These spectral detections are also valuable to probe *anti-trans*-MACRO and associated adducts to investigate reactions of MACRO with atmospheric species in laboratories, even though the limited detectivity of this technique and the small yield of MACRO might require a much greater proportion of the precursor.

## Methods

**Experimental**. The step-scan Fourier-transform infrared (FTIR) absorption technique is described in detail elsewhere[18,19]. A White cell of effective path length 3.6 m (base length 15 cm) and volume ~1370 cm³ was installed on the external port of the spectrometer to serve as a reactor and an absorption cell. A KrF excimer laser (248 nm, 6–11 Hz, ≈ 230 mJ pulse⁻¹, beam size 1.2 × 9.3 cm²) was employed to photodissociate either pure (E)-CH₂IC(CH₃)CHI (**1a**) or a mixture of (E)-CH₂IC(CH₃)CHI (**1a**) and (Z)-CH₂IC(CH₃)CHI (**1b**), denoted (**1**). The photolysis laser beam was multiply reflected between a pair of external laser mirrors and propagated sideways, nearly perpendicular to the IR beams in the White cell.

The IR probe light from the FTIR spectrometer was detected with a HgCdTe detector at 77 K; the signal was sent to an external 14-bit digitizer with a temporal resolution 4 ns. Period 40 µs (10,000 data points) was typically covered. In some cases, an internal 24-bit digitizer (temporal resolution 12.5 µs) was used to cover a longer period with an improved SNR. We employed appropriate optical filters to limit the spectral region so as to perform undersampling to decrease the data-acquisition time. For spectral range 753–1504 cm⁻¹ at instrumental resolution 1 cm⁻¹, 1523 scan steps (each averaged with 6–11 laser shots) were completed in ≈50 min. The spectral width (full width at half maximum) after apodization with the Blackman-Harris 3-term function is 1.28 times the listed instrumental

resolution. In all, 3-7 spectra accumulated under similar conditions were averaged to yield a spectrum with a satisfactory SNR.

The liquid sample of (**1**) was placed in a dark flask at 298 K; a stream of gaseous N₂ or O₂ was passed over the sample to carry the vapor into the reactor. The partial pressures of (**1**) were estimated with Beer's law using the observed integrated absorbance of IR bands in regions 1362–1412, 1272–1301, 759–791 cm⁻¹, and calculated IR intensities. The average photolysis fraction of (**1**) was estimated to be typically ~19% according to its decrease in infrared absorbance. The decrease of the precursor upon irradiation was estimated to be (8.9-11.1) × 10¹³ molecule cm⁻³. The flow rates were $F_{N_2} ≈ 21.4$ STP cm³ s⁻¹ (STP denotes standard temperature 273 K and pressure 1 atm) or $F_{O_2} ≈ 21.1$-73.9 STP cm³ s⁻¹. Partial pressures were $P_{CH_2IC(CH_3)CHI} ≈ 32$-55 mTorr, $P_{N_2} = 20.0$ Torr, or $P_{O_2} = 21.0$-346 Torr. (E)-CH₂IC(CH₃)CHI (**1a**) (>95%, Accela ChemBio), 1:1 mixture of (E)-/(Z)-CH₂IC(CH₃)CHI (**1**) (>97%, Accela ChemBio, the ratio of conformation was determined with NMR), N₂ (99.9995%, Chiah-Lung), and O₂ (99.99%, Chiah-Lung) were used as received. In this paper, we denote the mixture of conformers (E)-/(Z)-CH₂IC(CH₃)CHI as CH₂IC(CH₃)CHI or (**1**); the mixture is significantly more economical than (**1a**).

**Computational**. Quantum-chemical calculations were performed with the Gaussian 16 program suite[34]. The equilibrium geometry, rotational parameters, harmonic vibrational wavenumbers, and IR intensities of all conformers of precursor (**1**), isomers of CH₂C(CH₃)CHI (**2**) and CH₂IC(CH₃)CH (**6**), MACRO (**3**), dioxole, and the iodoperoxy adducts CH₂C(CH₃)CHIOO (**4**) and (CHI)C(CH₃)CH₂OO (**5**) were computed with the B3LYP density-functional theory (DFT), which uses Becke's three-parameter hybrid exchange functional with a correlation functional of Lee et al.[35–37]. The anharmonic vibrations were calculated for isomers of MACRO with a second-order perturbation approach using an effective finite-difference evaluation of the third and semi-diagonal fourth derivatives; as MACRO contains no I atom, anharmonic vibrational calculations for these species are more practical. In both methods, the standard Dunning's correlation-consistent basis set augmented with diffuse functions, aug-cc-pVTZ, was used[38,39]. For the iodine atom, the additional

**Table 4 Comparison of observed vibrational wavenumbers and IR intensities of $CH_2C(CH_3)CHIOO$ (4a) and $(CHI)C(CH_3)CH_2OO$ (5a) in region 800–1500 $cm^{-1}$ with those calculated with the B3LYP/aug-cc-pVTZ-pp method.**

| Mode | $CH_2C(CH_3)CHIOO$ (4a) | | | | $(CHI)C(CH_3)CH_2OO$ (5a) | | | |
| | Experiment | | Calculation | | Experiment | | Calculation | |
| | $\nu/cm^{-1}$ | Intensity[a] | $\nu/cm^{-1b}$ | Intensity[c] | $\nu/cm^{-1}$ | Intensity[a] | $\nu/cm^{-1b}$ | Intensity[c] |
|---|---|---|---|---|---|---|---|---|
| $\nu_8$ | | | 1463 | 17.5 | | | 1453 | 10.3 |
| $\nu_9$ | | | 1444 | 8.7 | | | 1450 | 1.9 |
| $\nu_{10}$ | | | 1420 | 2.9 | | | 1441 | 4.9 |
| $\nu_{11}$ | | | 1384 | 10.1 | | | 1385 | 9.0 |
| $\nu_{12}$ | | | 1299 | 5.6 | 1333 | 63 | 1332 | 49.9 |
| $\nu_{13}$ | | | 1267 | 3.1 | 1243 | 100 | 1280 | 74.4 |
| $\nu_{14}$ | | | 1144 | 9.3 | | | 1192 | 8.4 |
| $\nu_{15}$ | 1116 | 100 | 1135 | 36.0 | | | 1153 | 16.6 |
| $\nu_{16}$ | | | 1054 | 0.3 | | | 1147 | 2.6 |
| $\nu_{17}$ | 1031 | 50 | 1022 | 20.8 | | | 1049 | 2.0 |
| $\nu_{18}$ | | | 978 | 9.7 | 990 | 63 | 1034 | 29.0 |
| $\nu_{19}$ | 914 | 65 | 937 | 32.4 | | | 950 | 0.3 |
| $\nu_{20}$ | 888[d] | 75 | 902 | 58.7 | 886[d] | 79 | 889 | 17.3 |
| $\nu_{21}$ | | | 837 | 9.7 | | | 842 | 3.3 |

[a]Percentage of IR intensity relative to the most intense bands near 1116 and 1243 $cm^{-1}$, respectively.
[b]Harmonic vibrational wavenumber $x$ scaled according to $0.9683\,x + 11.5$; see text.
[c]In unit km $mol^{-1}$.
[d]Bands of $CH_2C(CH_3)CHIOO$ (4a) and $(CHI)C(CH_3)CH_2OO$ (5a) overlap.

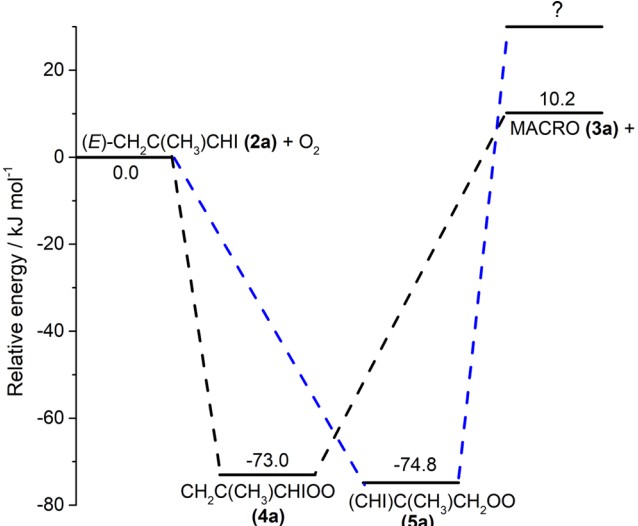

**Fig. 10 Potential energy schematics showing possible reactions of (E)-$CH_2C(CH_3)CHI$ (2a) with $O_2$.** Relative energies (kJ $mol^{-1}$) are computed with the CCSD(T)/aug-cc-pVTZ-pp//B3LYP/aug-cc-pVTZ-pp method and corrected with zero-point vibrational energy (ZPVE) from the harmonic vibrational wavenumbers calculated with the B3LYP/aug-cc-pVTZ-pp method.

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

pseudopotential, indicated as pp, was implemented[40]. In some cases, the geometries obtained from the B3LYP/aug-cc-pVTZ method were used for calculations of single-point electronic energies with the coupled-cluster single-double and perturbative triple, CCSD(T), method[41]. All energies were corrected for zero-point vibrational energy (ZPVE), which was taken from the harmonic vibrational energies calculated at the B3LYP level.

## Data availability

The data supporting the findings of this study are available within the paper and its Supplementary Information. All other relevant data are available from the authors upon reasonable request.

15. Vansco, M. F. et al. Synthesis, electronic spectroscopy, and photochemistry of methacrolein oxide: a four-carbon unsaturated Criegee intermediate from isoprene ozonolysis. *J. Am. Chem. Soc.* **141**, 15058–15069 (2019).

16. Lin, Y.-H., Yin, C., Takahashi, K. & Lin, J. J.-M. Surprisingly long lifetime of methacrolein oxide, an isoprene derived Criegee intermediate, under humid conditions. *Commun. Chem.* **4**, 12 (2020).

17. Barber, V. P. et al. Four-carbon Criegee intermediate from isoprene ozonolysis: methyl vinyl ketone oxide synthesis, infrared spectrum, and OH production. *J. Am. Chem. Soc.* **140**, 10866–10880 (2018).

18. Huang, Y.-H., Chen, J.-D., Hsu, K.-H., Chu, L.-K. & Lee, Y.-P. Transient infrared absorption spectra of reaction intermediates detected with a step-scan Fourier-transform infrared spectrometer. *J. Chin. Chem. Soc.* **61**, 47–58 (2014).

19. Su, Y.-T., Huang, Y.-H., Witek, H. A. & Lee, Y.-P. Infrared absorption spectrum of the simplest Criegee intermediate CH₂OO. *Science* **340**, 174–176 (2013).

20. Huang, Y.-H., Li, J., Guo, H. & Lee, Y.-P. Infrared spectrum of the simplest Criegee intermediate CH₂OO at resolution 0.25 cm⁻¹ and new assignments of bands 2ν₉ and ν₅. *J. Chem. Phys.* **142**, 214301 (2015).

21. Lin, H.-Y. et al. Infrared identification of the Criegee intermediates *syn-* and *anti-*CH₃CHOO, and their distinct conformation-dependent reactivity. *Nat. Commun.* **6**, 7012 (2015).

22. Wang, Y.-Y., Chung, C.-Y. & Lee, Y.-P. Infrared spectral identification of the Criegee intermediate (CH₃)₂COO. *J. Chem. Phys.* **145**, 154303 (2016).

23. Chung, C.-A. & Lee, Y.-P. Infrared characterization of formation and resonance stabilization of the Criegee intermediate methyl vinyl ketone oxide. *Commun. Chem.* **4**, 8 (2020).

24. Huang, Y.-H., Chen, L.-W. & Lee, Y.-P. Simultaneous infrared detection of the ICH₂OO radical and Criegee intermediate CH₂OO: The pressure dependence of the yield of CH₂OO in the reaction CH₂I + O₂. *J. Phys. Chem. Lett.* **6**, 4610–4615 (2015).

25. Su, Y.-T. et al. Extremely rapid self-reaction of the simplest Criegee intermediate CH₂OO and its implications in atmospheric chemistry. *Nat. Chem.* **6**, 477–483 (2014).

26. Wang, Y.-Y., Dash, M. R., Chung, C.-Y. & Lee, Y.-P. Detection of transient infrared absorption of SO₃ and 1,3,2-dioxathietane-2,2-dioxide [cyc-(CH2) O(SO₂)O] in the reaction CH₂OO + SO₂. *J. Chem. Phys.* **148**, 064301 (2018).

27. Chung, C.-A., Su, J.-W. & Lee, Y.-P. Detailed mechanism and kinetics of the reaction of Criegee intermediate CH₂OO with HCOOH investigated via infrared identification of conformers of hydroperoxymethyl formate and formic acid anhydride. *Phys. Chem. Chem. Phys.* **21**, 21445–21455 (2019).

28. Liang, W.-C., Luo, P.-L. & Lee, Y.-P. Infrared characterization of the products and the rate coefficient of the reaction between Criegee intermediate CH₂OO and HCl. *Phys. Chem. Chem. Phys.* **23**, 11082–11090 (2021).

29. Vansco, M. F. et al. Experimental evidence of dioxole unimolecular decay pathway for isoprene-derived Criegee intermediates. *J. Phys. Chem. A* **124**, 3542–3554 (2020).

30. Kuwata, K. T. & Valin, L. C. Quantum chemical and RRKM/master equation studies of isoprene ozonolysis: methacrolein and methacrolein oxide. *Chem. Phys. Lett.* **451**, 186–191 (2008).

31. Western, C. M. PGOPHER, A program for simulating rotational structure, version 10.1.181, University of Bristol, U.K. http://pgopher.chm.bris.ac.uk. (2018).

32. Caravana, R. L. et al. Direct kinetic measurements and theoretical predictions of an isoprene-derived Criegee intermediate. *Proc. Natl Acad. Sci. USA* **117**, 9733–9740 (2020).

33. Caravan, R. L., Vansco, M. F. & Lester, M. I. Open questions on the reactivity of Criegee intermediates. *Commun. Chem.* **4**, 44 (2020).

34. Frisch, M. J. et al. Gaussian 09, Revision A.02. Gaussian, Inc., Wallingford CT, USA (2016).

35. Becke, A. D. Density-functional thermochemistry. III. The role of exact exchange. *J. Chem. Phys.* **98**, 5648–5652 (1993).

36. Lee, C., Yang, W. & Parr, R. G. Development of the Colle-Salvetti correlation-energy formula into a functional of the electron density. *Phys. Rev. B* **37**, 785–789 (1988).

37. Miehlich, B., Savin, A., Stoll, H. & Preuss, H. Results obtained with the correlation energy density functionals of Becke and Lee, Yang and Parr. *Chem. Phys. Lett.* **157**, 200–206 (1989).

38. Dunning, T. H. Jr Gaussian basis sets for use in correlated molecular calculations. I. The atoms boron through neon and hydrogen. *J. Chem. Phys.* **90**, 1007–1023 (1989).

39. Woon, D. E. & Dunning, T. H. Jr Gaussian basis sets for use in correlated molecular calculations. III. The atoms aluminum through argon. *J. Chem. Phys.* **98**, 1358–1371 (1993).

40. Feller, D. The role of databases in support of computational chemistry calculations. *J. Comp. Chem.* **17**, 1571–1586 (1996).

41. Purvis, G. D. III & Bartlett, R. J. A full coupled-cluster singles and doubles model: the inclusion of disconnected triples. *J. Chem. Phys.* **76**, 1910–1918 (1982).

## Acknowledgements

This work was supported by the Ministry of Science and Technology, Taiwan (grants MOST110-2639-M-A49-001-ASP and MOST110-2634-F-009-026) and the Center for Emergent Functional Matter Science of National Chiao Tung University from The Featured Areas Research Center Program within the framework of the Higher Education Sprout Project by the Ministry of Education (MOE) in Taiwan. The National Center for High-Performance Computation provided computer time.

## Author contributions

J.-R.C. carried out some computations, all experiments, and initial analysis; J.-H.S. carried out some computations; Y.-P.L. formulated the research project, finalized the analysis, and wrote the manuscript with contributions from J.-R.C.

## Competing interests

The authors declare no competing interests.
