## [Peer Review File · Communications Chemistry]

Reviewers' comments:

Reviewer #1 (Remarks to the Author):

This paper presents a comprehensive study, using step-scan FTIR, of the mechanism of the formation of the MACRO carbonyl oxide, using a recently-developed synthetic approach. Using an approach that this group has established in previous papers, the authors report what is, to my knowledge, the first IR spectrum of the MACRO carbonyl oxide, as well as some of the coproducts of the diiodo synthetic approach, and elucidate key details of its formation mechanism. This result is novel and important, and is certainly suitable for publication in communications chemistry. However, I have several suggested revisions that I feel would improve the overall clarity and quality of the manuscript, as well as bring the conclusions in line with what the evidence in the paper shows, as enumerated below. I rate the overall amount of revision as somewhere between minor and major.

Throughout: I note that while the term "Criegee intermediate" is used throughout the literature, it is not the chemical name for these species, which are more correctly called carbonyl oxides. Rudolf Criegee, for whom Criegee intermediates are named, was a German scientist with ties to Nazism. If the authors feel strongly about using the term Criegee intermediate, it should not interfere with publication of this manuscript. However, I would request that they consider changing this terminology throughout the manuscript to carbonyl oxide.

Line 17: isoprene is the most abundantly-emitted non-methane volatile organic compound. Abundance is a function of both emissions and total reactivity- abundantly emitted is a more appropriate phrase here (and in line 28 as well).

Line 23: There is a grammatical issue with this sentence

Line 31: Given that OH is largely not produced by MACRO, this information is not particularly relevant to this paper.

Line 65-67: This sentence dramatically overstates the novelty of this work. UV-Vis spectroscopy of the MACRO carbonyl oxide has been published previously (Vansco et al., 2019). While certainly not as clearly conformer specific, the authors of this previous study do show that different conformers have different spectral behavior, and that the spectrum they observe is likely a combination of conformers. Further, the UV-Vis spectrum provides convincing support for resonance stabilization, because the UV absorption is shifted to longer wavelengths compared to alkyl-substituted carbonyl oxides. While the presentation of the mid-IR spectrum of MACRO is certainly novel and warrants publication, it is an overstatement to suggest that this is the first spectral evidence of resonance stabilization of this species.

Line 95: Given the wealth of computational data on MACRO that already exists, it would be valuable to briefly discuss how your calculation methodology and results compare to previous work. In particular, the Vansco et al. 2019 paper presents energy calculations of the MACRO conformers at a much higher level of theory than presented here.

Line 106: I suggest giving the energy difference here. In fact, throughout the text, the large number of referrals to supplementary information are quite distracting and make it difficult to follow the authors' arguments. I suggest the authors consider more carefully which evidence is really critical to making the arguments they present and put this in the main text. More detailed discussions of information in SI can go in the SI itself. Going back and forth between the SI and main text ought not to be necessary to follow the main crux of the authors argument, but it is here.

Line 132: Is this actually the relevant barrier? Or is it a barrier in the electronic excited state of 1? Scrambling of the conformers is believable, but I'm not sure this explanation (energy > barrier of

product isomerization in the ground electronic state) makes sense.

Line 135: At higher level of theory, this difference is even less, and so the behavior deviates even more substantially from Boltzmann behavior. This is quite an interesting result, since it hints at how the alternative carbonyl oxide synthetic method used in the lab might produce different chemistry from what one might observe in the atmosphere. Further discussion of possible explanations of this phenomenon would be extremely valuable.

Line 137: This sentence is quite unclear; I believe the authors mean dimerization reactions, not secondary reactions. It is unclear what is meant by "end product." More generally, I believe that most of the contents of this paragraph can be relocated to SI, and replaced with a single sentence summarizing the observation of the dimer as further evidence for breaking of the C-I allylic bond, since the primary evidence for this is in the previous paragraph. Alternatively, if the authors feel this evidence is critical to the paper, it would be appropriate to include a figure to this effect in the main text. Having to move back and forth between the main text and SI throughout this paragraph is extremely distracting to the reader.

Line 166: The rotational contours the authors observe are much less structured than predicted. Is this simply a signal/noise issue, or is there some actual spectroscopic reason for this? Either way, it warrants comment in the manuscript.

Line 171: It is very difficult to see how one can make an assignment based on relative intensity with only a single confidently assigned peak. By wavenumber alone, this could just as easily be the higher energy feature of the syn-cis conformer. They've also already stated that they observe non-Boltzmann behavior- the fact that this is the lowest energy conformer does not, on its own, mean that the spectrum can be attributed to it. Further, in the introduction and abstract they assert that they see the anti-trans conformer, and claim they have conformer-resolved detection. I do not find their spectral evidence to be convincing of either of these claims. In the absence of a distinct band contour, the authors' confidence in their assignment is, in my opinion overstated.

Further, there is a large feature observed at ca. 1150 cm⁻¹ in Figure 3 that is neither assigned nor discussed in the manuscript as far as I can tell.

Line 204: The MOs of MVKO and MACRO have been published previously and ought to be referenced here.

Line 209: Suggest framing this in terms of single and double bond character as opposed to a bond becoming single or double (we know that the reality is that both bonds have both single and double bond character).

Line 260: this methodological detail seems out of place here.

Line 290: MACRO is transient, so it must react away to form something else. Is there any evidence in your experimental data of what that product might be? Is there, perhaps, evidence of the self reaction product, or the dioxole at later times (as has recently been observed using photoionization mass spectrometry)? This would add quite a lot to the overall understanding of the reactivity of MACRO.

Figure 1: Formatting is strange in my copy, with faint boxes around molecules that look to be accidental

Figure 3: the simulated spectra are very strange looking- I suggest summing the simulations rather than overlaying them. The baselines underneath the features look quite strange.

Figure 4: There is something wrong with the labeling in this figure: What is fig 6(i and j)?

Reviewer #2 (Remarks to the Author):

This manuscript mainly described the detection of methacrolein oxide (MACRO) produced through UV irradiation of precursor $\text{CH}_2\text{C}(\text{CH}_3)\text{CHI}$ in excess O_2 by using a step-scan Fourier-transform infrared spectrometer. The method used by the author in this manuscript has been successfully used to detect the infrared spectrum of CH_2OO , CH_3CHOO , $(\text{CH}_3)_2\text{COO}$, MVKO. And the end products and some of the side reaction products were also identified. The method adopted by the author has certain credibility and the content of the article is substantial. Therefore, this article meets the submission requirements of this journal.

But after reviewing the manuscript, I have one question. In the study, the authors mainly identified the characteristic peaks of the main products in the reaction by comparing them with the simulated results of quantum chemical calculations. How does the author ensure the accuracy of the method used? Do you compare the results you have calculated with some existing standard spectra of iodide? I noticed that you used a formula $y = 0.9683x + 11.5$ to scale the harmonic vibrational wavenumbers of all species. What do these two values, 0.9683 and 11.5, mean?

In addition, there are still some mistakes need to be corrected in the manuscript.

Lines 43-44, The names of the compound do not correspond with the molecular formula. MACRO should be $\text{CH}_2\text{C}(\text{CH}_3)\text{CHOO}$. The chemical formula for iodide is also wrong.

Line 199, SNR appears for the first time without giving the full name. Line 327, signal-to-noise ratio (SNR) should be SNR.

The authors mark so many different compounds, including, 1a, 1b - 4c, 4d, 4e, 4f, etc. It's very laborious to read, so whether the author can give a schedule that corresponds the label to the molecular formula.

Reviewer #3 (Remarks to the Author):

This is a well performed study looking at the formation of methacrolein oxide.

The work reinforces the initial studies performed by the Lester group using Uv action spectroscopy and the Sandia group using multiplexed photoionization mass spectroscopy.

The novel aspect of the work is the use of the step -scan FTIR absorption spectroscopy. Using this technique the authors were able to show experimentally show for the first time that the precursor dissociated via the photolysis of the allylic C-I bond, which up to this point had just been assumed was how the Criegee intermediate was formed.

The authors claim that they have the first direct experimental of a conformer specific detection of MACRO. However, little discussion is presented on the multiplexed photoionization mass spectroscopy studies on MACRO. These studies also provide evidence of conformer specific identification of MACRO, via onset energy and shape of photoionization efficiency curves. Some more discussion of how the current study is better than the previous studies is required.

It would be nice for the authors to comment of whether this technique can be used for the study of conformer specific kinetics, or whether it is limited to the detection of conformers. Such studies, may be beyond the current paper, however can this technique be used for conformer specific MACRO studies?

Response/revisions to the reviewers' comments

We appreciate very much the valuable comments and suggestions from the reviewers; these really helped to improve the manuscript. Below are the detailed responses to the reviewer's comments on the manuscript. The reviewer comments are in black, our responses are listed in blue color after each comment, and the revised text are highlighted in yellow.

Reviewer #1 (Remarks to the Author):

This paper presents a comprehensive study, using step-scan FTIR, of the mechanism of the formation of the MACRO carbonyl oxide, using a recently-developed synthetic approach. Using an approach that this group has established in previous papers, the authors report what is, to my knowledge, the first IR spectrum of the MACRO carbonyl oxide, as well as some of the coproducts of the diiodo synthetic approach, and elucidate key details of its formation mechanism. This result is novel and important, and is certainly suitable for publication in communications chemistry. However, I have several suggested revisions that I feel would improve the overall clarity and quality of the manuscript, as well as bring the conclusions in line with what the evidence in the paper shows, as enumerated below. I rate the overall amount of revision as somewhere between minor and major.

Throughout: I note that while the term "Criegee intermediate" is used throughout the literature, it is not the chemical name for these species, which are more correctly called carbonyl oxides. Rudolf Criegee, for whom Criegee intermediates are named, was a German scientist with ties to Nazism. If the authors feel strongly about using the term Criegee intermediate, it should not interfere with publication of this manuscript. However, I would request that they consider changing this terminology throughout the manuscript to carbonyl oxide.

Response/Revision: We followed the reviewers suggestion and changed the name to carbonyl oxide, except the first appearance. The title was also changed.

Line 17: isoprene is the most abundantly-emitted non-methane volatile organic compound. Abundance is a function of both emissions and total reactivity- abundantly emitted is a more appropriate phrase here (and in line 28 as well).

Response/Revision: revised.

Line 23: There is a grammatical issue with this sentence

Response/Revision: The sentence has been revised to read: "Upon UV irradiation of (1) and O₂ near 21 Torr, *anti-trans*-MACRO (3a) was observed to have an intense OO-stretching absorption band near 917 cm⁻¹, much greater than those of *syn*-CH₃CHOO and (CH₃)₂COO, supporting a stronger O–O bond in MACRO because of resonance stabilization."

Line 31: Given that OH is largely not produced by MACRO, this information is not particularly relevant to this paper.

Response/Revision: The sentence and references “Some OH was predicted to be produced from the reaction isoprene + O₃⁵, with a yield about 25 %^{3,6,7}.” have been removed.

Line 65-67: This sentence dramatically overstates the novelty of this work. UV-Vis spectroscopy of the MACRO carbonyl oxide has been published previously (Vansco et al., 2019). While certainly not as clearly conformer specific, the authors of this previous study do show that different conformers have different spectral behavior, and that the spectrum they observe is likely a combination of conformers. Further, the UV-Vis spectrum provides convincing support for resonance stabilization, because the UV absorption is shifted to longer wavelengths compared to alkyl-substituted carbonyl oxides. While the presentation of the mid-IR spectrum of MACRO is certainly novel and warrants publication, it is an overstatement to suggest that this is the first spectral evidence of resonance stabilization of this species.

Response/Revision: In their UV spectrum, Vansco and coworkers only summed the predicted spectra of all four conformers (assuming equal population) and compared with observed spectrum; from the observed spectrum, they could not distinguish the contributions of each conformer. As for the resonance stabilization, we agree that the red-shift supports the resonance stabilization. To avoid confusion and argument, we have delete the sentence “To our knowledge, neither a conformation-specific spectral probe nor a direct spectral support for the resonance stabilization of MACRO (3) has been reported.”

Line 95: Given the wealth of computational data on MACRO that already exists, it would be valuable to briefly discuss how your calculation methodology and results compare to previous work. In particular, the Vansco et al. 2019 paper presents energy calculations of the MACRO conformers at a much higher level of theory than presented here.

Response/Revision: (1) We have added one sentence at the beginning of the quantum-chemical calculations section (page 4) to read: “Although Vansco *et al.* **Error!** **Bookmark not defined.** has reported high-level calculations for conformers of MACRO, we performed calculations at the B3LYP/aug-cc-pVTZ level of theory mainly for predictions of vibrational wavenumbers and IR intensities of various conformers of MACRO and other associated species.”

(2) Actually, we did compare the energies of the conformers of MACRO on current page 10, the last paragraph in the section “Infrared spectrum of carbonyl oxide *anti-trans*-MACRO (3a)” and also in Fig. 1. To make it more clearly, we also revised the last sentence of the paragraph in (1) on page 5 to read: “Relative energies of conformers are also listed in these figures and those of MACRO are compared with high-level calculations by Vansco *et al.*”.

Line 106: I suggest giving the energy difference here. In fact, throughout the text, the large number of referrals to supplementary information are quite distracting and make it difficult to follow the authors’ arguments. I suggest the authors consider more carefully which evidence is really critical to making the arguments they present and put this in the main text. More detailed discussions of information in SI can go in the SI itself. Going

back and forth between the SI and main text ought not to be necessary to follow the main crux of the authors argument, but it is here.

Response/Revision: (1) We have revised the sentence to read: “the (*E*)-conformer (**1a**) is predicted to have energy 1.6 kJ mol^{-1} greater than the (*Z*)-conformer (**1b**) at the B3LYP/aug-cc-pVTZ level of theory.” (2) We agree. We have moved three critical figures in SI (Original Figs. 6, 10, and 14) to the main text as Figs. 2, 6, and 8 (but also kept them in SI for easy referral in the discussion in SI) so that the readers do not have to go back and forth between SI and the main text; corresponding changes are highlighted with yellow.

Line 132: Is this actually the relevant barrier? Or is it a barrier in the electronic excited state of 1? Scrambling of the conformers is believable, but I’m not sure this explanation (energy > barrier of product isomerization in the ground electronic state) makes sense.

Response/Revision: The barrier is the electronic ground states of (**2a**) and (**2b**). We are assuming that most excessive energy remains in the radical part because the I atom has only translational energy. Actually, in the sentence, we only provided this background information to show that energy might not be a problem, but did not claim that this is the reason.

Line 135: At higher level of theory, this difference is even less, and so the behavior deviates even more substantially from Boltzmann behavior. This is quite an interesting result, since it hints at how the alternative carbonyl oxide synthetic method used in the lab might produce different chemistry from what one might observe in the atmosphere. Further discussion of possible explanations of this phenomenon would be extremely valuable.

Response/Revision: It is difficult for us to determine the relative populations of (*E*)- and (*Z*)-conformers of the radical (**2**) because the predicted spectra are similar except in region $1300\text{--}1350 \text{ cm}^{-1}$ and the predicted IR intensity might have a large error. To clarify this, we revised the sentence to: “but the observed spectra seem to show a contribution of (**2b**) small than the predicted ratio if one considers that a doublet of similar intensity was predicted for the (*Z*)-conformer in region $1300\text{--}1350 \text{ cm}^{-1}$ (Fig. 3c), but only one significant feature was observed. We are, however, uncertain about this population ratio because the predicted IR intensities might have large errors.”

Line 137: This sentence is quite unclear; I believe the authors mean dimerization reactions, not secondary reactions. It is unclear what is meant by “end product.” More generally, I believe that most of the contents of this paragraph can be relocated to SI, and replaced with a single sentence summarizing the observation of the dimer as further evidence for breaking of the C-I allylic bond, since the primary evidence for this is in the previous paragraph. Alternatively, if the authors feel this evidence is critical to the paper, it would be appropriate to include a figure to this effect in the main text. Having to move back and forth between the main text and SI throughout this paragraph is extremely distracting to the reader.

Response/Revision: Yes, we meant dimerization reactions, which are also one kind of secondary reactions. The wording “end products” means species that was produced at a

later time and remained nearly unchanged except for being pumped out. To clarify this, we have revised the sentence to be: “The product of dimerization of iodo-radicals (**2**) was also observed at a later period.” Also, in Conclusion, we revised the sentence to be “...(**7**), produced from the dimerization reaction of (**2**).” We feel that this information is important because this type of reaction product was never observed previously in experiments of other source radicals of carbonyl oxides, so, following the suggestion of the reviewer, we moved Supplementary Figs. 7 and 8 to the main text as Figs. 4 and 5.

Line 166: The rotational contours the authors observe are much less structured than predicted. Is this simply a signal/noise issue, or is there some actual spectroscopic reason for this? Either way, it warrants comment in the manuscript.

Response/Revision: We have added two sentences to comment on this difference as : ” The observed band near 917 cm^{-1} appears to have insignificant Q-branch and a broader rotational contour as compared with that simulated with the PGOPHER program, presumably because of the contribution of hot bands from excited states of the low-energy vibrational modes. Similar broadening was observed for carbonyl oxides containing a methyl rotor, $\text{CH}_3\text{CHOO}^{21}$, $(\text{CH}_3)_2\text{COO}^{22}$, and MVKO^{23} .” in the following paragraph on page 9.

Line 171: It is very difficult to see how one can make an assignment based on relative intensity with only a single confidently assigned peak. By wavenumber alone, this could just as easily be the higher energy feature of the syn-cis conformer. They’ve also already stated that they observe non-Boltzmann behavior- the fact that this is the lowest energy conformer does not, on its own, mean that the spectrum can be attributed to it. Further, in the introduction and abstract they assert that they see the anti-trans conformer, and claim they have conformer-resolved detection. I do not find their spectral evidence to be convincing of either of these claims. In the absence of a distinct band contour, the authors’ confidence in their assignment is, in my opinion overstated.

Further, there is a large feature observed at ca. 1150 cm^{-1} in Figure 3 that is neither assigned nor discussed in the manuscript as far as I can tell.

Response/Revision: (1) Spectral assignments without high resolution relies on pattern recognition. We have revised the paragraph to explain it more clearly to read: “Observed intense band B_1 near 917 cm^{-1} matches best with the spectrum simulated for *anti-trans*-MACRO (**3a**) in terms of vibrational wavenumbers and relative intensities. The OO-stretching (ν_{15}) band of (**3a**) was predicted at 944 cm^{-1} and to have the largest IR intensity (201 km mol^{-1}); other bands in region $850\text{--}1450\text{ cm}^{-1}$ have IR intensities less than 26 km mol^{-1} . For the four conformers, only (**3a**) was predicted to have a single intense band in this region; others were predicted to have more than two bands with comparable intensities. Furthermore, two *c*-type bands were predicted for modes ν_{24} and ν_{25} of (**3a**) near 950 and 924 cm^{-1} ; they might correspond to the small narrow features near 921 and 902 cm^{-1} , indicated with arrows in Fig. 3a; other conformers were predicted to have only one *c*-type bands near 900 cm^{-1} .” on page 9. (2) The band near 1150 cm^{-1} was due to interference of the intense absorption of the precursor and it does not appear to have the same intensity correlation with the main band near 917 cm^{-1} (See

Supplementary Figures 9 and 10); we marked these interference regions with grey. The assignment of *syn-cis*-MACRO is tentative, as stated in the text.

Line 204: The MOs of MVKO and MACRO have been published previously and ought to be referenced here.

Response/Revision: The MO that we presented (HOMO, HOMO-6, HOMO-8) are different from those by Vansco et al (HOMO-5 to LUMO+3) because we only wanted to show orbitals delocalized over the CCCOO skeleton. We added a note to refer to their work on page 11: “Furthermore, the molecular orbitals of MACRO and MVKO also show delocalization of π -electron densities over the CCCOO skeleton, as shown in Supplementary Fig. 11c for node = 0-2; a more complete set of molecular orbitals have been reported by Vansco *et al.* **Error! Bookmark not defined.**”.

Line 209: Suggest framing this in terms of single and double bond character as opposed to a bond becoming single or double (we know that the reality is that both bonds have both single and double bond character).

Response/Revision: We have revised the sentence to “In this hyper-conjugation structure, the O-O bond has single-bond character, whereas the adjacent C=O bond has double-bond character;”

Line 260: this methodological detail seems out of place here.

Response/Revision: We have deleted “(4 ns, 12-bit)”.

Line 290: MACRO is transient, so it must react away to form something else. Is there any evidence in your experimental data of what that product might be? Is there, perhaps, evidence of the self reaction product, or the dioxole at later times (as has recently been observed using photoionization mass spectrometry)? This would add quite a lot to the overall understanding of the reactivity of MACRO.

Response/Revision: As shown in (present) Supplementary Figs. 8c and 12d, the major product is methacrolein; this might be the produced from the self-reaction of MACRO. There are also some weaker broad features that we cannot identify at this moment. We described this on page 8 as “Processing observed difference spectra recorded with an internal ADC (Figs. 6a-c) by removing contributions of (2) and end product methacrolein and adding back the loss of the precursor yielded Figs. 6d-f; the regions in which the absorption of the precursor might interfere are marked with grey rectangles. Further processing by taking out contributions of other stable products (Fig. 6f) produced “cleaner” spectra, as shown in Figs. 6g and 6h.”

Figure 1: Formatting is strange in my copy, with faint boxes around molecules that look to be accidental

Response/Revision: Our figure in the merged pdf file seems to be fine. This might happen when the figure is printed on a dark background. We have fixed this problem.

Figure 3: the simulated spectra are very strange looking- I suggest summing the

simulations rather than overlaying them. The baselines underneath the features look quite strange.

Response/Revision: We intentionally not to sum them because the predicted wavenumbers are not exact; summing up all (overlapping) bands are sometimes misleading. Keeping each individual band is easier for the readers to know the contours of each band, especially for overlapped ones. The baseline was meant to be a guide for the contours of weak bands.

Figure 4: There is something wrong with the labeling in this figure: What is fig 6(i and j)?

Response/Revision: We forgot to change the figure numbers after we moved some figures to the supporting information. We have corrected them. Now they should be Fig. 8i and 8j.

Reviewer #2 (Remarks to the Author):

This manuscript mainly described the detection of methacrolein oxide (MACRO) produced through UV irradiation of precursor $\text{CH}_2\text{IC}(\text{CH}_3)\text{CHI}$ in excess O_2 by using a step-scan Fourier-transform infrared spectrometer. The method used by the author in this manuscript has been successfully used to detect the infrared spectrum of CH_2OO , CH_3CHOO , $(\text{CH}_3)_2\text{COO}$, MVKO. And the end products and some of the side reaction products were also identified. The method adopted by the author has certain credibility and the content of the article is substantial. Therefore, this article meets the submission requirements of this journal.

But after reviewing the manuscript, I have one question. In the study, the authors mainly identified the characteristic peaks of the main products in the reaction by comparing them with the simulated results of quantum chemical calculations. How does the author ensure the accuracy of the method used? Do you compare the results you have calculated with some existing standard spectra of iodide? I noticed that you used a formula $y = 0.9683x + 11.5$ to scale the harmonic vibrational wavenumbers of all species. What do these two values, 0.9683 and 11.5, mean?

Response/Revision: As stated on page 5, we compared the the observed bands of precursor (**1a**) with computed harmonic vibrational wavenumbers and fit them with a linear equation; the values 0.9683 and 11.5 are the slope and the intercept of the fitting, respectively. Most people used only the scaling factor without intercept, but we found that including the intercept gives better fit. To show the accuracy, we added one sentence as “The average absolute deviations of scaled harmonic vibrational wavenumbers of (**1a**) from observed wavenumbers is $7.5 \pm 6.4 \text{ cm}^{-1}$; the error represents one standard deviation in fitting.” on page 6.

In addition, there are still some mistakes need to be corrected in the manuscript. Lines 43-44, The names of the compound do not correspond with the molecular formula. MACRO should be $\text{CH}_2\text{C}(\text{CH}_3)\text{CHOO}$. The chemical formula for iodide is also wrong.

Response/Revision: Thanks for pointing out this error. We have correct the formula of MACRO (5 places) and the iodide ($\text{CH}_2\text{C}(\text{CH}_3)\text{CHI}_2$).

Line 199, SNR appears for the first time without giving the full name. Line 327, signal-to-noise ratio (SNR) should be SNR.

Response/Revision: We have corrected this.

The authors mark so many different compounds, including, 1a, 1b - 4c, 4d, 4e, 4f, etc. It's very laborious to read, so whether the author can give a schedule that corresponds the label to the molecular formula.

Response/Revision: The label and the formula of the main species, conformers of (1), (2), and (3), and single structures of (4) and (5) are actually presented in Fig. 1; conformers of (4), (5), and (6), which were not distinguished in most cases, are presented in Supplementary Fig. 3, 4, and 1, respectively. We have revised the sentence at the first line of page 4 to make this clear: "A detailed reaction scheme appears in Fig. 1; the chemical formula and labels of key species are also presented."

Reviewer #3 (Remarks to the Author):

This is a well performed study looking at the formation of methacrolein oxide.

The work reinforces the initial studies performed by the Lester group using Uv action spectroscopy and the Sandia group using multiplexed photoionization mass spectroscopy.

The novel aspect of the work is the use of the step -scan FTIR absorption spectroscopy. Using this technique the authors were able to show experimentally show for the first time that the precursor dissociated via the photolysis of the allylic C-I bond, which up to this point had just been assumed was how the Criegee intermediate was formed.

The authors claim that they have the first direct experimental of a conformer specific detection of MACRO. However, little discussion is presented on the multiplexed photoionization mass spectroscopy studies on MACRO. These studies also provide evidence of conformer specific identification of MACRO, via onset energy and shape of photoionization efficiency curves. Some more discussion of how the current study is better than the previous studies is required.

Response/Revision: To our knowledge, the work on the multiplexed photoionization mass spectral studies of MACRO has not been published. If the reviewer would enlighten us which paper it is, we would really appreciate it.

It would be nice for the authors to comment of whether this technique can be used for the study of conformer specific kinetics, or whether it is limited to the detection of

conformers. Such studies, may be beyond the current paper, however can this technique be used for conformer specific MACRO studies?

Response/Revision: Yes. In principle, this technique can be applied to investigate the kinetics associated with *anti-trans*-MACRO. However, the “conformation-specific” wording might lead the readers to think that we can detect all four conformers of MACRO, so we have changed the title to “Reaction mechanism and infrared spectra of methacrolein oxide *anti-trans*-CH₂C(CH₃)CHO and associated precursor and adduct radicals” and take out the wording of “conformation-specific”. We stated in Conclusion that “These spectral detections are also valuable to probe *anti-trans*-MACRO and associated adducts to investigate reactions of MACRO with atmospheric species in laboratories, even though the limited detectivity of this technique and the small yield of MACRO might require a much greater proportion of the precursor.”

REVIEWERS' COMMENTS:

Reviewer #1 (Remarks to the Author):

The authors have adequately addressed my concerns- I have no further revisions and recommend the manuscript be published as is.

Reviewer #2 (Remarks to the Author):

The authors had revised their manuscript carefully, now it is acceptable as it is.